# M4U: Evaluating Multilingual Understanding and Reasoning for Large Multimodal Models

## Abstract

Multilingual capability is an essential aspect for large multimodal models, since they are usually deployed across various countries and languages. However, most existing benchmarks for multilingual multimodal reasoning struggle to differentiate between models of varying performance; even language models without visual capabilities can easily achieve high scores. This leaves a comprehensive evaluation of leading multilingual multimodal models largely unexplored. In this work, we introduce M4U, a novel and challenging benchmark for assessing the capability of multi-discipline multilingual multimodal understanding and reasoning. M4U contains 8,931 samples covering 64 disciplines across 16 subfields in Science, Engineering, and Healthcare in Chinese, English, and German. Using M4U, we conduct extensive evaluations of 21 leading Large Multimodal Models (LMMs) and Large Language Models (LLMs) with external tools. The evaluation results show that the state-of-the-art model, GPT-4o, achieves only 47.6% average accuracy on M4U. Additionally, we observe that the leading LMMs exhibit significant language preferences. Our in-depth analysis indicates that leading LMMs, including GPT-4o, suffer performance degradation when prompted with cross-lingual multimodal questions, such as images with key textual information in Chinese while the question is in German. We believe that M4U can serve as a crucial tool for systematically evaluating LMMs based on their multilingual multimodal reasoning capabilities and monitoring their development.

## 1 Introduction

Multimodal reasoning is an essential aspect of human-level intelligence. AI systems with strong multimodal reasoning capabilities have extensive applications, including automatic scientific discovery, autonomous driving, and healthcare. The rapid advancements in Large Language Models (LLMs) (OpenAI, 2023; Touvron et al., 2023; Jiang et al., 2023) have led to the development of Large Multimodal Models (LMMs) (Anil et al., 2023; Liu et al., 2024; Lu et al., 2024a; AI, 2024), which demonstrate remarkable performance across a broad range of tasks, such as image captioning and visual question answering. Numerous benchmarks have been established to comprehensively evaluate these leading LMMs in real-world scenarios (Liu et al., 2023; Li et al., 2023a;d; Koh et al., 2024; Fu et al., 2023). Unlike perceptual tasks (Goyal et al., 2017; Chen et al., 2015), multimodal reasoning tasks, including mathematical reasoning (Lu et al., 2024b) and scientific question answering (Lu et al., 2022; Kembhavi et al., 2016), present significant challenges for neural models. These tasks necessitate an understanding of domain-specific knowledge and the ability to perform complex logical reasoning alongside visual content. Additionally, multilingual capability is crucial for real-world applications, as these models are typically deployed across various countries and languages.

Many datasets are curated to evaluate the capability of multilingual multimodal reasoning. However, the multimodal component of existing benchmarks (Zhang et al., 2023b; Das et al., 2024; Wang et al., 2023b) is limited in scale. We observe that the current data on multilingual multimodal reasoning suffers from language disparities in task complexity. For instance, the multimodal part of M3Exam (Zhang et al., 2023b) contains 61% high-difficulty questions in English, but only 23% high-difficulty questions in Chinese. Furthermore, the existing benchmark struggles to differentiate between models of varying multimodal capabilities. As shown in Table 2, without any visual information, the multilingual LLM, Qwen-1.5-14B Chat, easily achieves high scores: 66.4% and 56.0% accuracy on the Chinese and English sections of M3Exam, respectively. Consequently, a

| Statistics | Values |
|---|---|
| Languages | CN, EN, DE |
| Total questions | 8,931 |
| Total disciplines / subfields | 64 / 16 |
| Total image types | 13 |
| Image in the question | 8,271 |
|    Image at the beginning | 6,205 |
|    Image in the middle | 1,321 |
|    Image at the end | 745 |
| Image in the options | 660 |
| Single / multiple image(s) | 8,199 / 732 |
| Maximum question length | 279 |
| Maximum option length | 63 |
| Average question length | 33.2 |
| Average option length | 6.1 |

Figure 1: Key statistics of M4U dataset. M4U covers a wide scope of tasks from Science, Engineering and Health in Chinese, English and German, and supports the interleaved vision-language documents.

systematic evaluation of multilingual multimodal understanding and reasoning for leading models remains largely unexplored.

To advance the development of multilingual LMMs, we introduce M4U, a novel and challenging benchmark for evaluating foundational models on expert-level multilingual multimodal understanding and reasoning. Specifically, we assembled a team of over 10 college and graduate students to collect a high-quality dataset and assess its difficulty and correctness. As shown in Figure 1, M4U consists of 8,931 multiple-choice questions covering 64 disciplines across 16 subfields in Science, Engineering, and Healthcare. To minimize the risk of data contamination, samples are collected from college exams and quizzes from online video lectures. Additionally, a significant portion (35%) of the questions in M4U are written by our team based on textbooks. Figure 2 illustrates an example from the Chemistry-Inorganic part of M4U, demonstrating that our dataset requires expert-level multimodal reasoning and multilingual capability.

With M4U, we conduct a comprehensive evaluation, both quantitative and qualitative, on the zero-shot performance of 17 leading LMMs and 4 LLMs. Furthermore, we assess the performance of the LMMs with chain-of-thought prompting (Wei et al., 2022; Zhang et al., 2023c) and the LLMs with external tools, such as a powerful captioning model. As shown in Table 3 in §3.3, the most advanced model, GPT-4o (OpenAI, 2024a), achieves only 47.6% average accuracy with zero-shot prompting on the M4U dataset, demonstrating the significant challenge M4U poses for existing models. Additionally, we observe significant language preferences among the leading LMMs: InstructBLIP Vicuna-7B achieves 29.8% accuracy on the English section, but only 13.7% and 19.7% accuracy on the Chinese and German sections, respectively. Further results (§3.4) indicate that leading LMMs suffer performance degradation when prompted with cross-lingual multimodal questions, such as images with key textual information in Chinese while the question is in German. Our in-depth analysis (§4) reveals that the errors of GPT-4V(ision) are mainly due to limited perception ability, domain-specific knowledge, and reasoning. These findings demonstrate that LMMs still have significant room for improvement, particularly in multilingual multimodal reasoning.

## 2 THE M4U BENCHMARK

### 2.1 OVERVIEW

In this section, we introduce M4U, a novel and challenging benchmark for assessing the multilingual multimodal understanding and reasoning of foundational models. To investigate whether differences exist in the multimodal reasoning capabilities of LMMs across different languages, we first construct

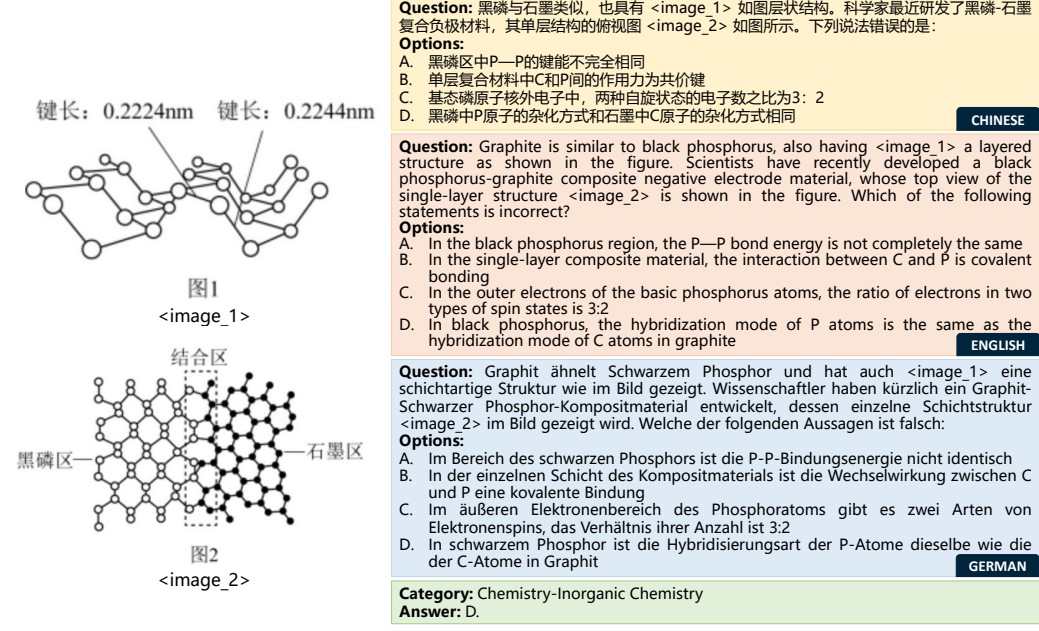

Figure 2: An example from the Chemistry-Inorganic of M4U dataset. The sample contains multiple images, and has multilingual contents in the question and images.

Table 1: Comparison between M4U and the existing benchmarks. *We report the size of test set of the multimodal part for the benchmarks.

| Benchmark | Multilingual | Multimodal | Size* | Difficulty | Fields |
|---|---|---|---|---|---|
| CMMLU (Li et al., 2023b) | ✗ | ✗ | - | ★★★ | STEM, Humanities, etc. |
| C-Eval (Huang et al., 2023) | ✗ | ✗ | - | ★★★ | STEM, Humanities, etc. |
| MMLU (Hendrycks et al., 2021) | ✗ | ✗ | - | ★★★ | STEM, Humanities, etc. |
| MathVista (Lu et al., 2024b) | ✗ | ✓ | 6,141 | ★★★ | Mathematics |
| CMMMU (Zhang et al., 2024) | ✗ | ✓ | 12,012 | ★★★ | Art, STEM, Humanities, etc. |
| MMMU (Yue et al., 2023) | ✗ | ✓ | 11,550 | ★★★ | Art, STEM, Humanities, etc. |
| MGSM (Shi et al., 2022) | ✓ | ✗ | - | ★★★ | Mathematics |
| EXAMS-V (Das et al., 2024) | ✓ | ✓ | 1,221 | ★★ | STEM, Humanities, etc. |
| M3Exam (Zhang et al., 2023b) | ✓ | ✓ | 2,816 | ★ | STEM, Humanities, etc. |
| M4U (ours) | ✓ | ✓ | 8,931 | ★★★★ | STEM, Healthcare |

the Chinese part of M4U and then translate it into English and German. This approach ensures that the domain-specific knowledge and reasoning abilities tested in different languages remain consistent. Specifically, we assemble a team of over 10 college students to collect questions from the Internet, textbooks, online video lectures, and college exams. Subsequently, a team of graduate students from related majors assessed the quality of the curated questions. Following this, we utilize GPT-4 Turbo (`gpt-4-turbo-preview`) to translate the questions into other languages, and then manually checked the quality of the translated questions.

The key statistics of M4U are detailed in Figure 1. M4U contains 8,931 multiple-choices questions, covering 64 subjects of 16 fields from Science (36.4%), Engineering (28.7%) and Health (34.9%) in Chinese, English and German. Different from the prior work (Zhang et al., 2023b), M4U includes interleaved image-text documents: 8.2% of the questions have multiple images, while the images of 14.8% and 7.4% of the questions are placed in the middle of question and the options, respectively. The average length of questions and options is 33.2 words and 6.1 words. The image sources of M4U cover 13 categories in different scenarios, e.g., diagrams, technical blueprints and medical images.

Table 2: The zero-shot accuracy of the multilingual LLM, Qwen-1.5-14B-Chat (Bai et al., 2023a), on the multimodal part of M3Exam and M4U dataset. We provide the scores of random choices in blue as the reference baseline. The higher scores of the text-only LLM indicate that the multimodal benchmark requires less visual efforts.

| Benchmark | Chinese↓ | English↓ | German↓ | Average↓ |
|---|---|---|---|---|
| M3Exam (Zhang et al., 2023b) | 66.4 (25.9) | 56.0 (25.0) | - | 61.2 (25.5) |
| M4U (ours) | 28.0 (25.9) | 19.7 (25.9) | 27.6 (25.9) | 25.1 (25.9) |

We present a comparison of M4U with existing benchmarks in Table 1. Unlike MMMU (Yue et al., 2023) and CMMMU (Zhang et al., 2024), our dataset focuses on the evaluation of multilingual multimodal reasoning. Furthermore, M4U is larger and has a more balanced distribution of difficulty across different languages compared to M3Exam (Zhang et al., 2023b). This ensures a fair comparison of models' capabilities in multimodal reasoning within multilingual scenarios. More importantly, we implement strict collection guidelines and quality control measures to minimize the risk of data contamination. To quantitatively measure the difficulty on visual capability for these benchmarks, we use the scores of LLMs without any visual information as a reference. Higher scores indicate that the benchmark is less effective at differentiating between models of varying multimodal performance. As shown in Table 2, without any visual information, Qwen-1.5-14B Chat easily achieves 66.4% and 56.0% accuracy on the Chinese and English parts of M3Exam, respectively, while it only achieves 25.1% average accuracy on our dataset. This suggests that M4U is more challenging and less exposed to the training corpus of LLMs.

## 2.2 DATA COLLECTION

**Data sources.** Following MMMU (Yue et al., 2023), we go through the educational programs of top universities, then select 64 subjects of 16 subfields from Science, Engineering and Healthcare whose applications highly rely on visual information. We recruit a team of over 10 college students to collect multiple-choices questions from public available sources. To minimize the risk of data contamination for foundation models, different from M3Exam (Zhang et al., 2023b), we do not include the samples from the official exam papers, e.g., National Postgraduate Entrance Examination and national professional exams. Although these resources usually have higher quality and are well organized, they are also easy to be curated for the training of LLMs. Therefore, we carefully select the data sources for M4U: most questions of our dataset are collected from the quizzes of online video lectures and college exams in PDF documents. Further 35% of questions are written by our team according to the textbooks. As shown in Table 3, the state-of-the-art open-source multilingual LLM, Qwen-1.5-14B Chat (Bai et al., 2023a), only has 25.1% average accuracy on M4U dataset without any visual information. This proves that our data is less exposed in the training of language models.

**Data processing.** The primary sources of M4U include college exams, the quizzes of online video lectures and the written questions. Most of college exams are uploaded by their students as images or scanned PDF documents, while the quizzes of online video lectures can be taken as the screenshot. We first adopt the OCR tools to convert these images into plain texts, then manually correct the potential errors of OCR results. Besides, we also write a large portion (35%) of questions according to the textbooks. For the mathematical formulas and the chemical structures, we require the annotators to convert them into LaTeX format. Since the samples of M4U may include multiple images in the questions or options, the annotators also annotate the location and type of each image (e.g. tables, blueprints and medical images).

After collecting the data, we design a two-stage post-processing pipeline to further improve the quality of M4U. We first design the guidelines to allow each annotator to score the collected samples from three dimensions: image quality, question description quality and the difficulty of visual understanding, and filter out the questions with average scores lower than 2.0. We present the distribution of image resolution for M4U in Figure 7 of Appendix B. The detailed guidelines are summarized in the Appendix A.2. Then we recruit a team of graduate students of related major to assess the difficulty and quality of the curated questions. We further filter out the questions with

the minimum visual efforts and the wrong answer. After that, we use GPT-4 Turbo to translate the Chinese part of M4U to English and German. Then the annotators will check and correct the potential errors introduced by machine translation.

## 2.3 EVALUATION

We evaluate the zero-shot performance of 17 leading LMMs of different scales on M4U. The models are required to follow the instruction to directly generate the predicted option for each question. To minimize the format discrepancy between training and evaluation, we handle models that support interleaved image-text documents by inserting the visual tokens of each image into the corresponding position as in training. For models that only support image-text pairs as input, we place all visual tokens at the beginning of the sentence and use annotated positions (e.g., <image_1>, <image_2>) to refer to each image. Furthermore, we also evaluate the performance of various LMMs with chain-of-thought prompting (Wei et al., 2022; Zhang et al., 2023c) and LLMs equipped with detailed visual captions.

## 3 EXPERIMENTS

### 3.1 SETUP

We evaluate the performance of zero-shot learning for various LMMs and LLMs of different scales across different languages on M4U dataset. The models are prompted to directly generate the option's letter. Further we also evaluate the performance of the LMMs with chain-of-thought prompting (Wei et al., 2022; Zhang et al., 2023c): the models should first generate the rationale for the question and the options, then give the predicted option. For reference, we add the baseline of Random choices: we randomly select an option, and use the average accuracy of 30 runs with different seeds. We provide more details about the instruction prompt in Appendix A.3. We adopt NVIDIA A40 for evaluations.

### 3.2 MODELS

**LMMs.**  For the open-source models, we select VisualGLM (Du et al., 2022), Ying-VLM (Li et al., 2023c), InstructBLIP series (Dai et al., 2023), InternLM-XComposer (Zhang et al., 2023a), CogVLM-Chat (Wang et al., 2023a), Qwen-VL-Chat (Bai et al., 2023b), Yi-VL-series (AI, 2024), DeepSeek-VL (Lu et al., 2024a) and LLaVA-NeXT series (Liu et al., 2024). For closed source models, we choose Gemini 1.0 Pro (Anil et al., 2023), GPT-4V(ision) (OpenAI, 2023) and GPT-4o (OpenAI, 2024a) using the provided API, `gemini-pro-vision`, `gpt-4-vision-preview` and `gpt-4o`, respectively. As for the augmented LMMs, we evaluate the performance of Gemini 1.0 Pro and GPT-4V with the chain-of-thought prompting (Wei et al., 2022; Zhang et al., 2023c).

**LLMs.**  We select Mistral-Instruct-v0.2-7B (Jiang et al., 2023), Qwen-1.5-7B-Chat, Qwen-1.5-14B-Chat (Bai et al., 2023a) and Gemini 1.0 Pro (`gemini-pro`) for the open and closed source LLMs. We use Gemini 1.0 Pro (`gemini-pro-vision`) to generate the detailed caption in Chinese, English and German for each image. The visual captions are placed at the beginning of the prompt. The annotated image positions are used to refer each image.

### 3.3 MAIN RESULTS

In this section, we present the comprehensive evaluation results of 17 leading LMMs and 4 LLMs with different prompt strategies. Table 3 demonstrates the performance of various LMMs and LLMs across Chinese, English and German on M4U dataset.

For the text-only LLMs, we first only use the text part of question to prompt these models. As shown in Table 3, Qwen-1.5-14B Chat has only 25.1% average accuracy on M4U dataset, which is lower than 25.9% of random choices. It proves that M4U requires significant visual efforts to answer these questions. Further, we equip these LLMs with the detailed visual captions generated by Gemini 1.0 Pro. Qwen-1.5-14B Chat with additional captions outperforms itself without any visual information by a gain of 7.7%, and achieves 32.8% average accuracy, the highest scores among the baselines. Mistral-Instruct-v0.2-7B has 25.6% average accuracy, since it does not follow the instruction to

Table 3: The zero-shot accuracy of various LLMs, augmented LLMs and LMMs on M4U dataset. *CoT* is short for chain-of-thought prompting.

| Models | Size | Chinese↑ | English↑ | German↑ | Average↑ |
|---|---|---|---|---|---|
| Random choices | - | 25.9 | | | |
| *Large Language Models* | | | | | |
| Qwen-1.5-7B-Chat (Bai et al., 2023a) | 7B | 29.5 | 15.0 | 28.5 | 24.3 |
| Qwen-1.5-14B-Chat (Bai et al., 2023a) | 14B | 28.0 | 19.7 | 27.6 | 25.1 |
| *Augmented Large Language Models (+ Visual Caption)* | | | | | |
| Mistral-Instruct-v0.2-7B (Jiang et al., 2023) | 7B | 24.9 | 24.9 | 26.9 | 25.6 |
| Gemini 1.0 Pro (Anil et al., 2023) | - | 31.6 | 31.1 | 30.9 | 31.2 |
| Qwen-1.5-7B-Chat (Bai et al., 2023a) | 7B | 34.2 | 27.7 | 31.7 | 31.2 |
| Qwen-1.5-14B-Chat (Bai et al., 2023a) | 14B | 32.7 | 32.0 | 33.8 | 32.8 |
| *Large Multimodal Models* | | | | | |
| VisualGLM (Du et al., 2022) | 6B | 8.7 | 22.4 | 13.5 | 14.9 |
| Ying-VLM (Li et al., 2023c) | 13B | 22.3 | 11.2 | 15.6 | 16.4 |
| InstructBLIP-Vicuna-13B (Dai et al., 2023) | 13B | 10.5 | 23.4 | 18.6 | 17.5 |
| InstructBLIP-Vicuna-7B (Dai et al., 2023) | 7B | 13.7 | 28.1 | 19.7 | 20.5 |
| LLaVA-NeXT-Vicuna-7B (Liu et al., 2024) | 7B | 11.8 | 29.8 | 28.2 | 23.3 |
| LLaVA-NeXT-Vicuna-13B (Liu et al., 2024) | 13B | 21.9 | 30.9 | 29.3 | 27.4 |
| Qwen-VL-Chat (Bai et al., 2023b) | 7B | 29.7 | 29.9 | 27.1 | 28.9 |
| CogVLM-Chat (Wang et al., 2023a) | 7B | 28.9 | 30.2 | 28.5 | 29.2 |
| LLaVA-NeXT-Mistral-7B (Liu et al., 2024) | 7B | 28.2 | 30.6 | 29.4 | 29.4 |
| InternLM-XComposer (Zhang et al., 2023a) | 7B | 31.8 | 31.6 | 29.1 | 30.8 |
| DeepSeek-VL (Lu et al., 2024a) | 7B | 30.4 | 32.8 | 30.8 | 31.3 |
| Yi-VL-6B (AI, 2024) | 6B | 33.4 | 31.4 | 29.7 | 31.5 |
| Yi-VL-34B (AI, 2024) | 34B | 33.5 | 33.3 | 30.5 | 32.4 |
| Gemini 1.0 Pro (Anil et al., 2023) | - | 34.9 | 32.7 | 30.8 | 32.8 |
| LLaVA-NeXT-34B (Liu et al., 2024) | 34B | 38.5 | 36.2 | 35.2 | 36.6 |
| GPT-4V(ision) (OpenAI, 2023) | - | 39.7 | 39.4 | 37.3 | 38.8 |
| GPT-4o (OpenAI, 2024a) | - | **49.4** | **47.8** | **45.6** | **47.6** |
| *Augmented Large Multimodal Models* | | | | | |
| Gemini 1.0 Pro (Anil et al., 2023) + *CoT* | - | 34.4 | 34.2 | 33.9 | 34.2 |
| GPT-4V(ision) (OpenAI, 2023) + *CoT* | - | **43.9** | **43.6** | **40.3** | **42.6** |

generate the valid option. We observe that Mistral-Instruct-v0.2-7B tends to reject to give an answer when not being provided with enough visual information.

For LMMs, most of them do not have the satisfactory results on M4U dataset. The state-of-the-art model, GPT-4o, achieves only 47.6% average accuracy with zero-shot prompting. It indicates that M4U is quite challenging for the existing models, and the reasoning capability of the multimodal models still has much room for future improvement. We present the detailed results of GPT-4o across 64 disciplines in Figure 8 of Appendix B. With the powerful LLM, Nous-Hermes Yi-34B [1], LLaVA-NeXT-34B scores highest among the open-source LMMs, even significantly outperforms Gemini 1.0 Pro by a gain of 3.8% on average accuracy. As for augmented LMMs, chain-of-thought prompting further boosts the performance. GPT-4V with chain-of-thought prompting outperforms itself with zero-shot prompting by a gain of 3.8% on average accuracy. It demonstrates that explicitly generating the reasoning steps is also beneficial for complex multimodal reasoning.

Furthermore, we observe that the existing models has strong language preferences on multilingual multimodal reasoning tasks. InstructBLIP Vicuna-7B achieves 28.1% accuracy on the English part of M4U, while only has 13.7% and 19.7% accuracy on the Chinese and German part, respectively. For GPT-4V, the average accuracy on the Chinese and English is both 3% higher than it on the German. Besides, we observe that the effect of chain-of-thought prompting also differs across different languages. For instance, chain-of-thought improves the performance of Gemini 1.0 Pro on English and German part by a gain of 1.5% and 3.1% accuracy, while leads to a degradation of 0.5% on Chinese part. We argue that this results from the lack of the multilingual vision-language corpus

---

[1]https://huggingface.co/NousResearch/Nous-Hermes-2-Yi-34B

Table 4: The zero-shot accuracy of various LMMs, augmented LMMs on the cross-lingual set of M4U dataset. *CoT* is short for chain-of-thought prompting.

| Models | Size | Chinese↑ | English↑ | German↑ | Average↑ |
|---|---|---|---|---|---|
| DeepSeek-VL (Lu et al., 2024a) | 7B | 32.8 | 34.0 | 33.3 | 33.4 |
| Yi-VL-6B (AI, 2024) | 6B | 39.2 | 34.3 | 30.1 | 34.5 |
| Gemini 1.0 Pro (Anil et al., 2023) | - | 38.0 | 36.3 | 32.9 | 35.7 |
| Yi-VL-34B (AI, 2024) | 34B | 41.6 | 38.7 | 34.2 | 38.2 |
| LLaVA-NeXT-34B (Liu et al., 2024) | 34B | 44.6 | 40.9 | 36.1 | 40.5 |
| GPT-4V(ision) (OpenAI, 2023) | - | 45.3 | 41.2 | 38.2 | 41.6 |
| GPT-4o (OpenAI, 2024a) | - | **52.0** | **47.5** | **45.2** | **48.2** |
| Gemini 1.0 Pro (Anil et al., 2023) + *CoT* | - | 38.1 | 35.7 | 37.8 | 37.2 |
| GPT-4V(ision) (OpenAI, 2023) + *CoT* | - | **46.7** | **48.0** | **42.6** | **45.8** |

Table 5: The detailed results of different LMMs on Health, Science and Engineering of M4U dataset. Sci. and Eng. are short for Science and Engineering, respectively.

| Models | Chinese | | | English | | | German | | |
|---|---|---|---|---|---|---|---|---|---|
| | Health | Sci. | Eng. | Health | Sci. | Eng. | Health | Sci. | Eng. |
| Yi-VL-6B | 31.2 | 34.1 | 34.9 | 32.1 | 32.2 | 30.0 | 29.0 | 29.2 | 30.8 |
| DeepSeek-VL | **40.1** | 22.6 | 28.5 | **38.0** | 31.9 | 28.6 | 35.2 | 29.3 | 27.8 |
| Yi-34B | 32.9 | 34.1 | 33.6 | 34.0 | 33.2 | 32.6 | 29.4 | 30.2 | 32.0 |
| LLaVA-NeXT-34B | 38.1 | **40.4** | 37.0 | 37.2 | **36.8** | **34.7** | 36.9 | **34.2** | **34.5** |
| Gemini 1.0 Pro | 38.8 | 34.5 | 31.4 | 34.9 | 33.4 | 29.8 | 33.1 | 30.5 | 28.8 |
| + *Chain-of-thought prompting* | 37.8 | 33.3 | 32.6 | 38.8 | 33.3 | 30.6 | 37.8 | 32.2 | 31.8 |
| GPT-4V(ision) | 41.9 | 39.3 | 37.9 | 43.9 | 37.8 | 36.6 | 41.1 | 36.0 | 34.6 |
| + *Chain-of-thought prompting* | 43.9 | 46.2 | 41.7 | 45.8 | 43.3 | 41.9 | 42.5 | 39.1 | 39.3 |
| GPT-4o | **56.0** | **47.3** | **45.0** | **56.2** | **44.3** | **42.8** | **52.9** | **40.9** | **43.0** |

used for multimodal training, and the LLMs of these LMMs (e.g., Vicuna-7B, Vicuna-13B) do not well support the multilingual capability.

### 3.4 Cross-lingual Multimodal Evaluation

To measure the cross-lingual multimodal capability of the LMMs, we select a subset from M4U: the image of each sample in this subset contains the text that labels or explains the key concepts or objects in the picture, while the textual description of the question is multilingual. For example, as illustrated in Figure 2, the visual content contains the key text in Chinese that labels bond length between atoms and explains the single-layer structure of the material, and the question part is multilingual. The models are required to perform complex reasoning given the multilingual both textual and visual contents. The cross-lingual set contains 1,065, 417 and 531 samples from Science, Engineering and Healthcare, respectively, resulting in up to 2,013 samples in total.

We evaluate the performance of different LMMs on the cross-lingual set of M4U. As shown in Table 4, almost all models suffer from a degradation of performance when the image contains the key textual information in Chinese but the question is English or German. It shows that these models are short for following multilingual instructions to understand the visual contents with the textual information of another language. Furthermore, as for the augmented LMMs, we observe that the chain-of-thought prompting significantly improves the performance of GPT-4V on English and German. This is aligned with our previous evaluations on the full set of M4U in Table 3.

### 3.5 Fine-grained results

**Different Disciplines and Languages.** We present the detailed results of various LMMs on different fields of Chinese, English and German in Table 5. GPT-4o outperforms the other models by large improvements on all fields of all languages. For the open-source models, we observe that

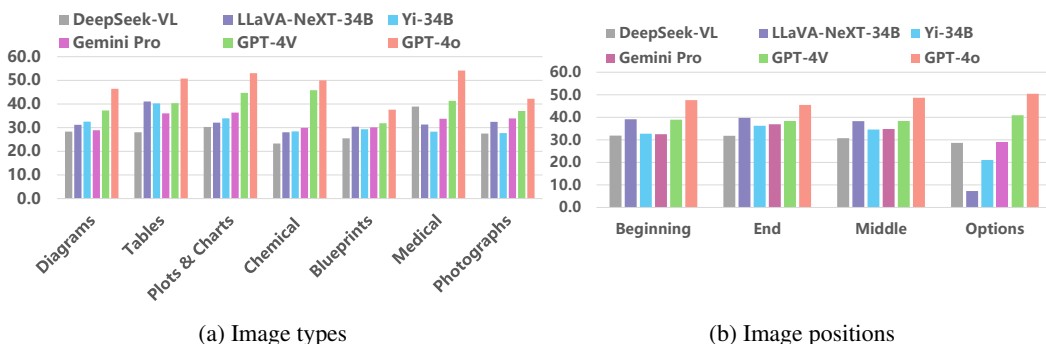

(a) Image types                    (b) Image positions

Figure 3: The zero-shot accuracy of different LMMs on different image types (Left) and positions (Right) on M4U dataset.

Table 6: The zero-shot accuracy of LMMs on the M4U-mini dataset.

| Models | English | Chinese | German | Japanese | Arabic | Thai | Avg. |
|---|---|---|---|---|---|---|---|
| DeepSeeK-VL-Chat | 35.4 | 33.6 | 35.0 | 32.1 | 24.8 | 25.4 | 31.0 |
| LLaVA-NeXT-34B | 44.1 | 44.2 | 39.0 | 36.0 | 11.4 | 34.0 | 34.8 |
| Gemini 1.5 Flash | 35.4 | 46.3 | **42.8** | 39.0 | 38.4 | 40.1 | 40.3 |
| GPT-4o | **44.9** | **53.7** | 42.4 | **49.1** | **45.2** | **48.8** | **47.3** |

LLaVA-NeXT-34B shows impressive results on scientific reasoning, and DeepSeek-VL demonstrates good performance on Health. Further, we observe that on Science, the chain-of-thought prompting significantly improves the performance of GPT-4V by a gain of over 6% accuracy in Chinese and English, while only boosts the performance by an improvement of 3.1% accuracy on German. The similar phenomenon also exists for Gemini 1.0 Pro. On Health part, Gemini 1.0 Pro with the chain-of-thought prompting outperforms it with zero-shot prompting by a gain of 4.9% and 4.7% on English and German, but it leads to a degradation of 1.0% accuracy on Chinese. These results show that the effect of the chain-of-thought prompting also differ from different languages.

**Different Image Types and Positions.** We demonstrate the visualization of the detailed results of various LMMs on different image types and positions in Figure 3. We reclassified 13 image types into 7 categories based on the style and application of the image. As shown in Figure 3a, GPT-4o shows impressive performance on the image type of "Plots & Charts" and "Medical" compared with the other models, but has unsatisfactory results on Blueprints. We argue that this is because the Blueprints contain many engineering sketches that require the capability of the fine-grained perception and domain-specific knowledge about engineering standards. M4U not only supports the image-text pairs as the input, but includes interleaved image-text documents. Thus, we conduct the analysis about the performance of the selected LMMs on different positions of the images. We divide these questions into four groups according to the image position: image at the beginning, end, middle of the question and in the options. As shown in Figure 3b, on the questions with images in the options, GPT-4o and GPT-4V outperform the other models by a large gain, and LLaVA-NeXT-34B performs poorly on this types of the questions. We argue that this is because the LLaVA-NeXT series are only trained with a high-quality corpus of image-text pairs. Instead DeepSeek-VL is pre-trained with a large mixture of image-text pairs and interleaved documents, and it does not suffer from a significant degradation of performance on the questions with images in the options.

### 3.6 EVALUATION ON M4U-MINI

To support more medium- or low-resource languages, we present M4U-mini, a tiny version of M4U with three additional languages (Japanese, Arabic, and Thai). We randomly select 5% of the test data and follow our processing pipeline to construct these parts. M4U-mini contains 1,076 samples of six languages. We evaluate the zero-shot performance of GPT-4o, Gemini-1.5-Flash, LLaVA-NeXT-34B and DeepSeeK-VL-Chat on M4U-mini. We present the results in Table 6. It demonstrates that GPT-

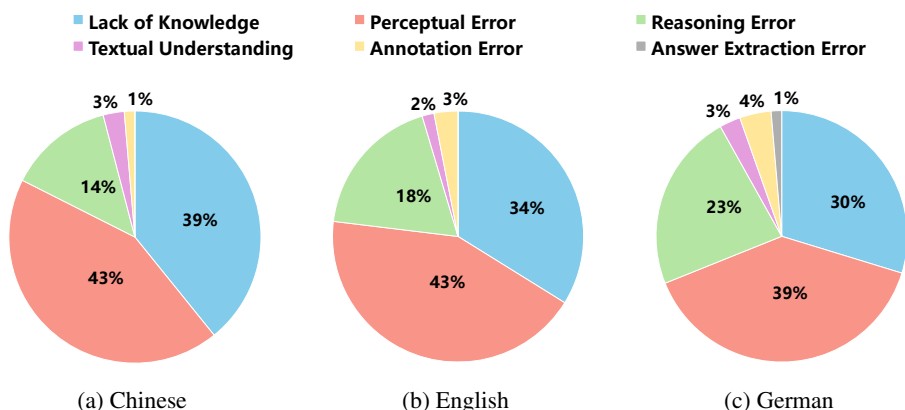

Figure 4: The distribution of the wrong cases of GPT-4V in different languages.

4o significantly outperforms LLaVA-NeXT-34B by a gain of 33.8% on Arabic part of M4U-mini. The open-sourced models, DeepSeeK-VL-Chat and LLaVA-NeXT-34B, both have performance degradation on medium- or low-resource languages, e.g., Arabic and Thai.

## 4 QUALITATIVE ANALYSIS

We conduct qualitative analysis for GPT-4V with the chain-of-thought prompting. Specifically, we randomly sample 75 questions (2.5%) from different disciplines of each language. In these instances, GPT-4V has errors in responses and analysis in at least one language. We analyze the cause of these wrong cases, and divided them into six categories: perceptual error, lack of knowledge, reasoning error, textual understanding, annotation error and answer extraction error.

The distribution of selected samples across different categories are illustrated in Figure 4. Perceptual error, lack of knowledge, and reasoning error account for the major causes of failed cases (96% in Chinese, 95% in English, and 92% in German). Regarding perceptual error illustrated in Figure 14, GPT-4V identified the position labeled 2 as the support above the compass, but it actually points to the compass needle. For reasoning error, as shown in Figure 26, GPT-4V only considers the power supply on the left and does not consider the power supply on the right. We observe that GPT-4V tends to exhibit lack of knowledge on the Chinese part of M4U, while reasoning errors are more likely to occur in German and English. More results can be found in the Appendix C.

**Perceptual Error.** Perceptual error is the most frequent error made by GPT-4V. It corresponds to the illusion phenomenon that occurs when extracting visual information from images provided by the questions. These kinds of hallucination could be divided in two main categories: visual information deficiency and misinterpretation. As presented in Figure 22, visual information deficiency occurs when GPT-4V overlooked crucial conditions and information provided in the image associated with the question, such as dimensions and scales annotated in engineering blueprints and numerical values provided in physics experiments thereby failing to complete the reasoning chain. Figure 16 shows a typical case for the visual information misinterpretation: the extracted information is complete but contains mistakes. A portion of these mistakes are common perceptual errors in OCR and visual localization.

**Lack of Knowledge.** We define the lack of knowledge as the model has factual misunderstanding about the key concepts in questions and provides erroneous premise to the reasoning process. Figure 24 illustrates that GPT-4V ignores that the amplitude needs to be judged on an equivalent basis to the same period, and as shown in Figure 25, the model equates the average kinetic energy of a molecule to the kinetic energy of a single molecule, overlooking key preconditions of physical laws.

**Reasoning Error.** The reasoning error is categorized as the mathematical miscalculations and logical errors in the analysis procedure, which often occur in subjects need numerical computations

and logical inference, such as math, physics, and electronics. As demonstrated in Figure 26, GPT-4V only considers the power supply on the left and does not consider the power supply on the right.

**Others.** Remaining error cases only occupy a small portion in selected cases, while depict long-tailed but various error reasons including textual misunderstanding, annotation error, and answer extraction error. Annotation error caused by typo or translation issues maintains less than 5% after manually checked by annotators.

## 5 RELATED WORK

Recent years have witnessed a trend towards large-scale multimodal pre-training, which aims to unify various vision-language tasks with a single model (Dai et al., 2023; Lu et al., 2024a; Bai et al., 2023b; OpenAI, 2023; 2024a). With the rapid progress of LMMs, previous benchmarks (e.g., VQA-v2 (Goyal et al., 2017), GQA (Hudson & Manning, 2019)) are insufficient to comprehensively evaluate the general multimodal capability of these models. Therefore, many datasets are curated for evaluate different aspects of multimodal capability, spanning from robustness and hallucination (e.g., POPE (Li et al., 2023d)) to general perception capability (e.g., MM-Vet (Yu et al., 2024) and MMBench (Liu et al., 2023)). As for multimodal reasoning tasks, MathVista (Lu et al., 2024b) presented a collection of diverse challenging mathematical and visual tasks. After that, instead focusing on the mathematical domain, MMMU (Yue et al., 2023) introduced a large-scale collection of more difficult expert-level problems that cover 30 different subjects. However, these benchmarks are primarily focused on English.

Multilingual capability is crucial for LLMs and LMMs. Many holistic evaluations have been conducted for LLMs, such as PAWS-X (Yang et al., 2019), XCOPA (Ponti et al., 2020), XStoryCloze (Lin et al., 2022), MGSM (Shi et al., 2022), MMMLU OpenAI (2024b). As for the evaluation of LMMs, M3Exam (Zhang et al., 2023b) collects the official exam papers of 9 different languages. However, they mainly focus on the evaluation of language capability. Despite M3Exam contains the samples with image as the input, its multimodal part is limited in scale. Besides, M3Exam struggles to differentiate between models of varying multimodal performance. As shown in Table 2, a language model without visual capabilities, Qwen-1.5-14B-Chat can easily achieve high scores. Different from previous works, M4U covers over 64 disciplines across Science, Engineering and Healthcare. We conduct the strict collection guidelines and quality control, which ensures that M4U requires significant visual efforts and domain-specific knowledge to perform multi-step reasoning.

## 6 CONCLUSION

In this work, we introduce M4U, a novel and challenging benchmark for evaluating the capability of multilingual multimodal understanding and reasoning. M4U contains 8,931 multiple-choice questions, covering 64 disciplines across 16 subfields in Science, Engineering, and Healthcare in Chinese, English, and German. Table 2 demonstrates that M4U requires significant visual efforts compared with M3Exam. As shown in Table 3, the state-of-the-art model, GPT-4o, achieves only 47.6% average accuracy with zero-shot prompting, indicating that M4U is quite challenging for existing models. Furthermore, we observe that the leading LMMs exhibit significant language preferences. These results demonstrate that there is still significant room for improvement in LMMs, especially in expert-level multilingual multimodal reasoning.

## 7 LIMITATION AND FUTURE WORK

Currently M4U focuses on the evaluation of science problems for multimodal reasoning. In the future, we aim to extend M4U to support more languages and investigate the performance of multilingual LMMs on questions associated with cultural backgrounds (e.g., history and politics). Additionally, we plan to include multilingual rationales for M4U to construct a fine-grained metric that considers the correctness of both reasoning steps and final predictions.

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

## A    DATASET DOCUMENTATION

### A.1    DATA SOURCES

M4U consists of 8,931 multiple-choices questions, covering 64 disciplines of 16 subfields from Science, Engineering and Healthcare. To minimize the risk of data contamination, the samples are collected from college exams, the quizzes of online video lectures. Further a large portion (35%) of the questions are written by our team according to the textbooks. The guidelines for annotators stress the importance of strictly following copyright and licensing rules from the original data sources, particularly avoiding materials from websites that prohibit copying and redistribution. If you come across any data samples that may violate copyright or licensing regulations, please inform us. Once verified, such samples will be promptly removed.

### A.2    FIRST-STAGE PROCESSING GUIDELINES

We summarize the detailed first-stage processing guidelines for the annotators. For each dimension, we require the annotator to score the sample following the below guidelines. The question with the higher scores indicates higher quality. We filter out the questions with average scores lower than 2.0. As for the image quality, the standard is:

- 0 score: The image is extremely blurry, difficult to recognize, or most of it is cropped, resulting in severe information loss.

- 1 score: The image is relatively blurry, details are hard to discern, or parts of the image are cropped, leading to some information loss.

- 2 score: The image is slightly blurry; most content is recognizable but details are unclear, or the image is slightly cropped, but most information is complete.

- 3 score: The image is mostly clear; all major content is recognizable, though some details may not be clear.

- 4 score: The image is clear; all content and details are easily recognizable with no apparent defects.

- 5 score: The image is very clear; details are excellently represented, complete without any cropping or obstructions, meeting or exceeding the expected quality standards.

The guideline for measuring the question description quality is:

- 0 score: The question is vague and completely unintelligible, with no clear intent.

- 1 score: The question statement is ambiguous, difficult to fully understand its intent, with multiple possible interpretations.

- 2 score: The question statement is basically clear, but there are some ambiguities or lack of rigor that need further clarification.

- 3 score: The question statement is clear, though there are some details that are not rigorous or there is slight ambiguity.

- 4 score: The question statement is both clear and rigorous, with details well handled, and only very minor issues present.

- 5 score: The question statement is extremely clear and rigorous, logical, without any ambiguity, fully meeting high standards.

The standard for measuring the difficulty of visual understanding is:

- 0 score: The question almost does not rely on visual ability, can be fully understood without any visual information.

- 1 score: The question does not completely rely on visual ability, both visual and non-visual information are balanced.

- 2 score: Although the question relies on visual ability, a considerable proportion of non-visual information assists understanding.

- 3 score: The question largely depends on visual ability, but some non-visual information is provided.
- 4 score: The question greatly depends on visual ability, with very little content provided by non-visual information.
- 5 score: The question completely depends on visual ability, without it, the content is incomprehensible.

Furthermore, we recruit a team of graduate student of related majors to access the difficulty and correctness for the questions. The team will filter out the questions with wrong answer or minor visual efforts.

## A.3 EVALUATION PROMPT

Figure 5 shows the prompt template used in the zero-shot evaluation of M4U. As for the chain-of-thought prompting, we first prompt the model to generate the rationale for the question and the options using the template shown in Figure 6, then append the generated rationale after the options. After that, the model are prompted to generate the predicted option.

```
{问题}
A. {选项_1}
B. {选项_2}
C. {选项_3}
D. {选项_4}
直接用给定选项的字母回答.

{Frage}
A. {Optionen_1}
B. {Optionen_2}
C. {Optionen_3}
D. {Optionen_4}
Antworten Sie direkt mit dem Buchstaben der gegebenen Optionen.

{Question}
A. {Options_1}
B. {Options_2}
C. {Options_3}
D. {Options_4}
Answer with the option's letter from the given choices directly.
```

Figure 5: The template of different languages used for the evaluation of the LLMs and LMMs.

```
{问题}
A. {选项_1}
B. {选项_2}
C. {选项_3}
D. {选项_4}
请分析题目和选项.

{Frage}
A. {Optionen_1}
B. {Optionen_2}
C. {Optionen_3}
D. {Optionen_4}
Bitte analysieren Sie die Frage und die Optionen.

{Question}
A. {Options_1}
B. {Options_2}
C. {Options_3}
D. {Options_4}
Please analyze the question and options.
```

Figure 6: The template of different languages used for the evaluation of the LMMs with the chain-of-thought prompting.

## B  MORE VISUALIZATIONS

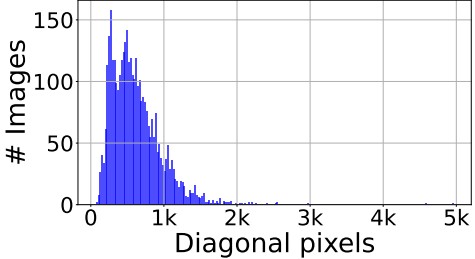

Figure 7: The distribution of image resolution for M4U dataset.

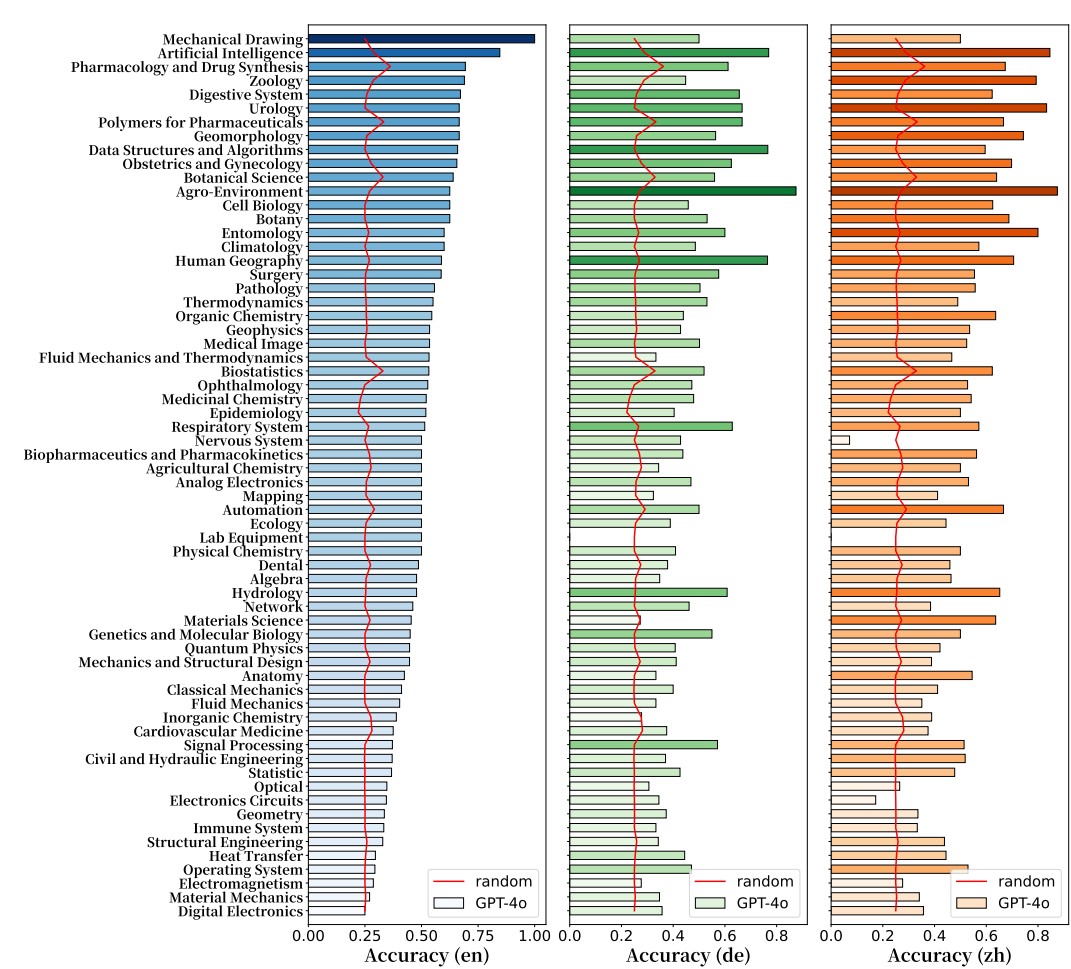

Figure 8: The zero-shot accuracy of GPT-4o across 64 subjects on M4U dataset.

## C    DETAILED RESULTS OF QUALITATIVE ANALYSIS

### C.1    CORRECT CASES

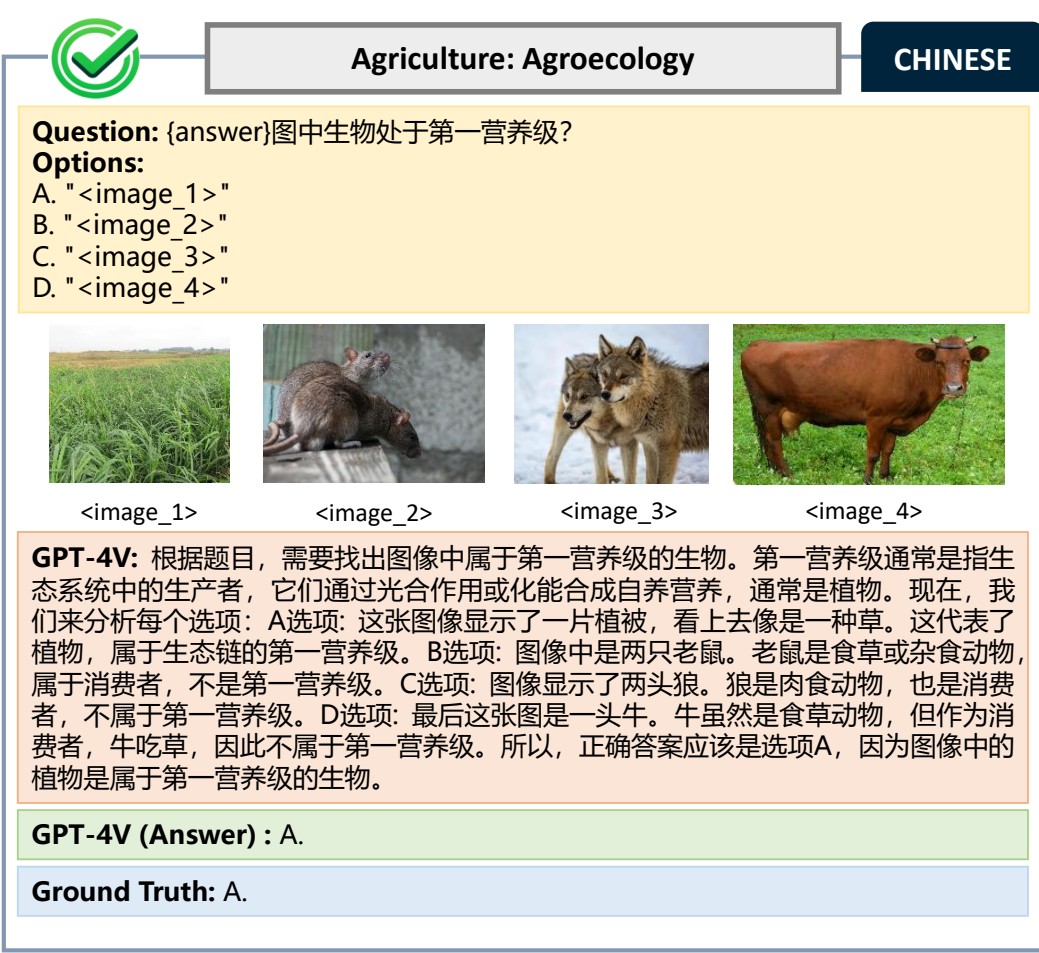

**Question:** {answer}图中生物处于第一营养级？
**Options:**
A. "<image_1>"
B. "<image_2>"
C. "<image_3>"
D. "<image_4>"

<image_1>    <image_2>    <image_3>    <image_4>

**GPT-4V:** 根据题目，需要找出图像中属于第一营养级的生物。第一营养级通常是指生态系统中的生产者，它们通过光合作用或化能合成自养营养，通常是植物。现在，我们来分析每个选项：A选项：这张图像显示了一片植被，看上去像是一种草。这代表了植物，属于生态链的第一营养级。B选项：图像中是两只老鼠。老鼠是食草或杂食动物，属于消费者，不是第一营养级。C选项：图像显示了两头狼。狼是肉食动物，也是消费者，不属于第一营养级。D选项：最后这张图是一头牛。牛虽然是食草动物，但作为消费者，牛吃草，因此不属于第一营养级。所以，正确答案应该是选项A，因为图像中的植物是属于第一营养级的生物。

**GPT-4V (Answer) :** A.

**Ground Truth:** A.

Figure 9: A sample correct case (subject: agriculture, language: Chinese).

918
919
920
921
922
923
924
925
926
927
928
929
930
931
932
933
934
935
936
937
938
939
940
941
942
943
944
945
946
947
948
949
950
951
952
953
954
955
956
957
958
959
960
961

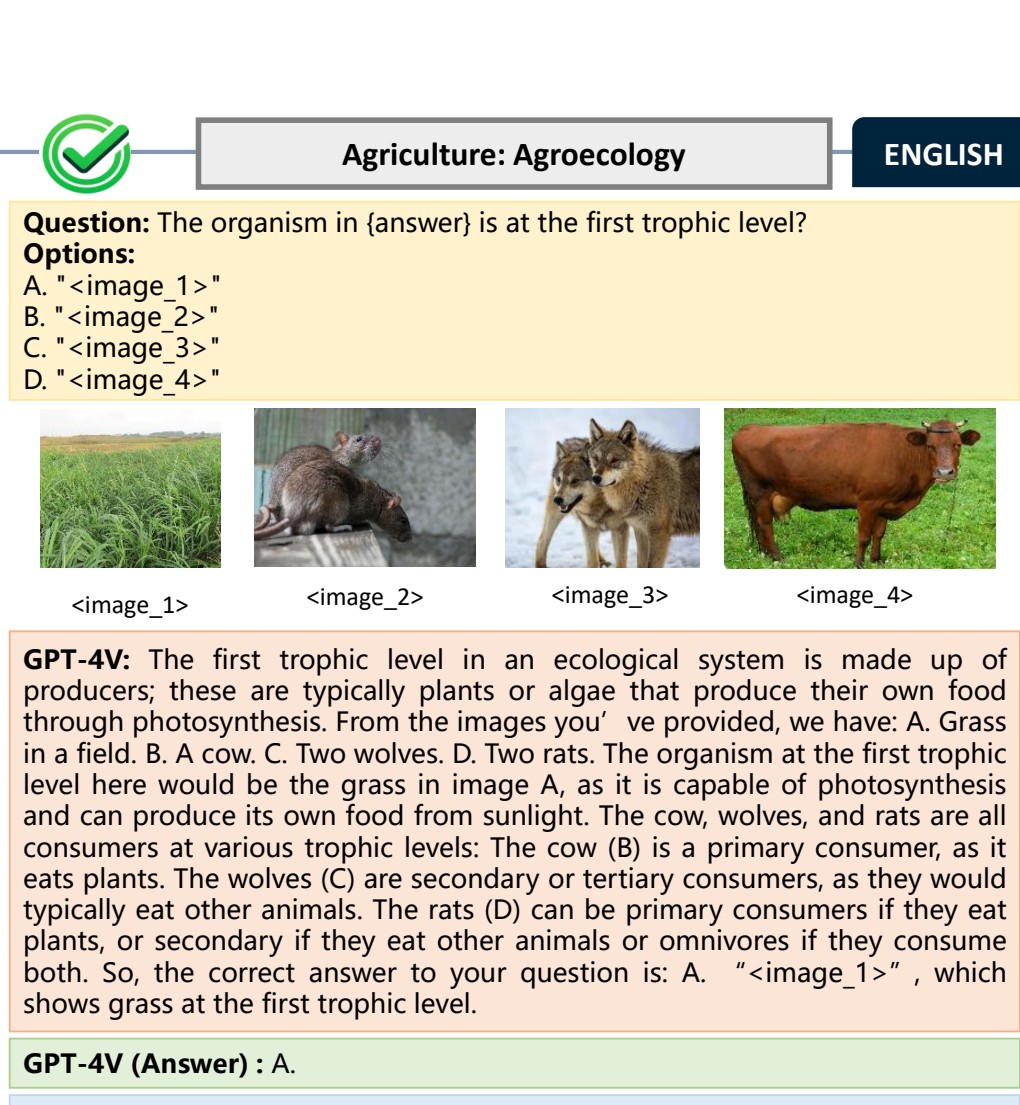

Figure 10: A sample correct case (subject: agriculture, language: English).

962
963
964
965
966
967
968
969
970
971

**Agriculture: Agroecology**    **GERMAN**

**Question:** {answer} im Bild befindet sich auf der ersten trophischen Ebene?
**Options:**
A. "<image_1>"
B. "<image_2>"
C. "<image_3>"
D. "<image_4>"

<image_1>   <image_2>   <image_3>   <image_4>

**GPT-4V:** In einem Ökosystem sind die verschiedenen trophischen Ebenen die Stufen der Nahrungskette, beginnend mit den Produzenten an der Basis, gefolgt von den verschiedenen Konsumenten. Hier eine kurze Erläuterung jeder trophischen Ebene: Die erste trophische Ebene wird von Primärproduzenten besetzt, das sind in der Regel Pflanzen oder Algen, die durch Photosynthese Energie aus Sonnenlicht in chemische Energie umwandeln. Die zweite trophische Ebene umfasst Primärkonsumenten oder Herbivore, die sich von den Primärproduzenten ernähren. Die dritte Ebene besteht aus Sekundärkonsumenten, typischerweise Fleischfresser, die sich von Herbivoren ernähren. Darüber hinaus gibt es weitere Ebenen mit Tertiärkonsumenten und so weiter, die sich von Sekundärkonsumenten ernähren. Von den Bildern, die Sie hochgeladen haben, wäre das Bild, das die Organismen der ersten trophischen Ebene zeigt (Primärproduzenten), das mit Pflanzen darauf, da sie die Energie direkt von der Sonne durch Photosynthese nutzen. Also, um Ihre Frage zu beantworten: A. "<image_1>": Pflanzen, gehören zu den Primärproduzenten. B. "<image_2>": Ratten, gehören zu den Konsumenten. C. "<image_3>": Wölfe, gehören zu den Konsumenten. D. "<image_4>": Eine Kuh, gehört zu den Konsumenten (Herbivoren). Die erste trophische Ebene im Bild befindet sich auf: A. "<image_1>", da dies das Bild mit den Pflanzen ist.

**GPT-4V (Answer) :** A.

**Ground Truth:** A.

Figure 11: A sample correct case (subject: agriculture, language: German).

**Physics: Classical Mechanics** **GERMAN**

**Question:** Nehmen Sie ein langes flexibles Seil, halten Sie ein Ende fest und ziehen Sie es gerade. Wenn Sie es dann kontinuierlich periodisch nach oben und unten schwingen lassen, können Sie eine Reihe von Wellen auf dem Seil sehen, wobei a, b, c Massenpunkte auf dem Seil sind. Im Moment, wenn die Welle gerade den Punkt c erreicht hat, ist die Wellenform auf dem Seil wie in der Abbildung <image_1> dargestellt. Welche der folgenden Aussagen ist korrekt?

**Options:**
A. Die Hand beginnt sich nach oben zu bewegen
B. Danach kehrt der Punkt a vor dem Punkt b zur Gleichgewichtslage zurück
C. Zu diesem Zeitpunkt ist die Geschwindigkeit des Punktes a größer als die Geschwindigkeit des Punktes b
D. Die Wellenform im Bild entspricht dem Zustand 1,5 Perioden nach Beginn der Schwingung der Wellenquelle

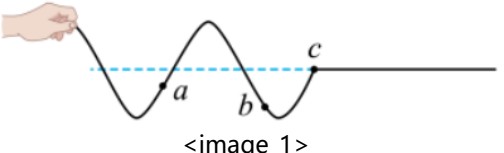

<image_1>

**GPT-4V:** Um die Frage zu beantworten, betrachten wir zunächst das Bild und die dargestellte Wellenform. Wir sehen einen Momentaufnahme des Seils, das eine stehende Welle ausbildet. Die Punkte a, b, und c sind Markierungspunkte auf dem Seil. Jetzt analysieren wir die Optionen: A. Die Hand beginnt sich nach oben zu bewegen. Ohne zusätzlichen Kontext ist es schwierig zu sagen, ob diese Aussage korrekt ist. Wir können aus dem Bild nicht direkt ableiten, in welche Richtung sich die Hand bewegt. Diese Aussage könnte man nur bestätigen, wenn man Informationen darüber hätte, welche Bewegung vor diesem Moment stattgefunden hat. B. Danach kehrt der Punkt a vor dem Punkt b zur Gleichgewichtslage zurück. Punkt b ist auf der Welle höher als Punkt a und wird daher eine längere Strecke zurücklegen, um zur Gleichgewichtslage zurückzukehren, als Punkt a. Das macht diese Aussage plausibel, aber wir brauchen mehr Kontext, um definitiv zu sein. C. Zu diesem Zeitpunkt ist die Geschwindigkeit des Punktes a größer als die Geschwindigkeit des Punktes b. Diese Aussage ist abhängig von der Position im Wellenzyklus. Da Punkt a näher an der Gleichgewichtslage ist, wo die Geschwindigkeit der schwingenden Punkte am größten ist, während Punkt b sich am Scheitelpunkt (einem Punkt maximaler Auslenkung der Welle) befindet, wo die Geschwindigkeit momentan null ist, ist diese Aussage wahrscheinlich korrekt. D. Die Wellenform im Bild entspricht dem Zustand 1,5 Perioden nach Beginn der Schwingung der Wellenquelle. Da wir nur ein Bild des Seils haben und keine Information über die Zeit, die vergangen ist, können wir nicht genau sagen, welcher Zustand dies in Bezug auf die Perioden der Welle darstellt. Es ist unmöglich, diese Aussage anhand des vorliegenden Bildes zu überprüfen. Nach dieser Analyse scheint Option C die plausibelste Aussage zu sein, basierend auf dem, was wir über die Dynamik von Wellen und Bewegungen von Punkten auf diesen Wellen wissen. In dem Moment, da Punkt b seinen Höhepunkt erreicht hat und seine Geschwindigkeit gleich null ist, müsste die Geschwindigkeit von Punkt a, der sich in der Nähe der Gleichgewichtslage befindet, größer sein. Antwort: C

**GPT-4V (Answer) :** C.

**Ground Truth:** C.

Figure 12: A sample correct case (subject: physics, language: German).

## C.2 PERCEPTUAL ERROR CASES

**Geography: Geophysics** — **CHINESE**

**Question:** 判断正误，日环食发生时，月球、地球和太阳三者间的位置关系正确的是 <image_1>：
**Options:**
A. 对
B. 错

<image_1>

**GPT-4V:** 这张图片描绘了太阳、地球和月球的相对位置。日环食发生时，月球位于地球和太阳之间，但由于月球距离地球较远，导致无法完全遮挡太阳，而是在太阳中心形成一个环状的光环。根据图片，太阳位于左侧，月球位在中间，地球位于右侧，这表明月球处在地球和太阳之间，这是正确的位置顺序，以形成日环食。因此，根据所提供的图片，该题目描述的月球、地球和太阳的位置关系正确。答案是：A

**GPT-4V (Answer) :** A.

**Error Category:** Perception Error
**Error Analysis:** There is a clear positional error in GPT's understanding of the image. The moon is depicted on the outermost edge in the image, but GPT mistakenly believes the moon is positioned between the Earth and the Sun. This erroneous analysis of relative positions leads to incorrect answers.

**Ground Truth:** B.

Figure 13: A sample perceptual error case (subject: geography, language: Chinese).

| ⊗ | **Architecture: Mapping** | **ENGLISH** |

**Question:** <image_1> The figure shows a compass, and the part labeled as 2 is called what?
**Options:**
A. Telescope
B. Support
A. Clamp screw
B. Magnetic needle

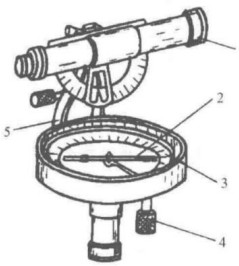

<image_1>

**GPT-4V:** To answer the question, we need to identify each labeled part of the compass based on its visual characteristics and typical functions in a compass: 1. Label 1 is pointing to a long cylindrical part, which appears to be a sighting device used to align the compass with a specific object or direction. This is typically known as the telescope or sighting scope. 2. Label 2 is pointing to the arm that supports the sighting device over the body of the compass. This is consistent with the general function of a support structure, which is to hold another part in position. 3. Label 3 is pointing to the main body of the compass, which seems to encase the compass's directional mechanisms and the glass through which one observes the magnetic needle. 4. Label 4 indicates a part at the bottom of the compass, which typically refers to the mechanism that allows the compass to be leveled or adjusted for stability, often involving screws or feet. 5. Label 5 points to the circular scale around the outside of the compass, likely used to measure angles or bearings. Now, let's evaluate each option with the above analysis: A. Telescope - While label 1 would be associated with the telescope, this is not the part marked as 2. B. Support - Considering the function of the part labeled 2, this option makes the most sense as it is the structure supporting the telescope. C. Clamp screw - This is typically a component used to secure something tightly in place, and the part labeled 2 does not seem to have a screw element visible, so this isn't the most appropriate choice. D. Magnetic needle - Label 3 points to the body of the compass where the magnetic needle would be located, not label 2. Based on the analysis, the most appropriate term for the part labeled as 2 would be the support. The correct answer is: B

**GPT-4V (Answer) :** B.

**Error Category:** Perceptual Error
**Error Analysis:** A typical identical and location error. GPT identified the position labeled 2 as the support above the compass, but it actually points to the compass needle.

**Ground Truth:** D.

Figure 14: A sample perceptual error case (subject: architecture, language: English).

| Math: Algebra | CHINESE |
| --- | --- |

**Question:** 设行列式<image_1>，则$D_1$的值为?
**Options:**
A. -15
B. -6
C. 6
D. 15

$$D=\begin{vmatrix} a_{11} & a_{12} & a_{13} \\ a_{21} & a_{22} & a_{23} \\ a_{31} & a_{32} & a_{33} \end{vmatrix}=3, \quad D_1=\begin{vmatrix} a_{11} & 5a_{11}+2a_{12} & a_{13} \\ a_{21} & 5a_{21}+2a_{22} & a_{23} \\ a_{31} & 5a_{31}+2a_{32} & a_{33} \end{vmatrix}$$

<image_1>

**GPT-4V:** 为了计算行列式$D_1$的值，我们可以利用行列式的性质。根据题目中已知的信息，D的值为-3。现在我们需要计算$D_1$的值。 $D_1$可以看作是通过将D的第二行的每个元素乘以5后再加上第一行的相应元素的两倍来得到的一个新行列式。这个变换可以分成两步：1. 将原行列式D的第二行的每个元素乘以5。2. 将原行列式D的第一行的每个元素乘以2后加到新的第二行对应的元素上。由于行列式的一个性质是当你将一个行（或列）乘以一个常数时，行列式的值也会乘以这个常数，所以在第一步，将第二行每个元素乘以5后， D的值会乘以5。因此，步骤1的行列式的值为 $5 \times (-3) = -15$。然而，在步骤2，我们只是将第一行的两倍加到第二行上，并没有改变行列式的值。在行列式中，你可以将一行的若干倍加到另一行而不改变行列式的值。因此，步骤2不会影响行列式的值。结合这两步，行列式$D_1$的值就是第一步的结果，即-15。所以答案是：A. -15

**GPT-4V (Answer) :** A.

**Error Category:** Perception Error
**Error Analysis:** GPT recognized '3' in an image as '-3'，which is a typical OCR error.

**Ground Truth:** C.

Figure 15: A sample perceptual error case (subject: math, language: Chinese).

**Math: Algebra** ENGLISH

**Question:** Given the determinant <image_1>, the value of $D_1$ is ?
**Options:**
A. –15
B. -6
C. 6
D. 15

$$D = \begin{vmatrix} a_{11} & a_{12} & a_{13} \\ a_{21} & a_{22} & a_{23} \\ a_{31} & a_{32} & a_{33} \end{vmatrix} = 3, \quad D_1 = \begin{vmatrix} a_{11} & 5a_{11} + 2a_{12} & a_{13} \\ a_{21} & 5a_{21} + 2a_{22} & a_{23} \\ a_{31} & 5a_{31} + 2a_{32} & a_{33} \end{vmatrix}$$

<image_1>

**GPT-4V:** To find the value of $D_1$, we need to compare the determinants $D$ and $D_1$. In the given image, there is a determinant $D$ and a modified determinant $D_1$. The determinant $D$ is a 3x3. The determinant $D_1$ is the same 3x3 matrix, but with its second column transformed by multiplying each element by 5 first and then adding the first column to it. Now, recall that if we multiply a column in a determinant by a constant, the value of the determinant is multiplied by that constant. Additionally, if we add a multiple of one column to another column in a determinant, the value of the determinant remains unchanged. The second column of $D_1$ can be thought of as being formed by multiplying the first column of $D$ by 5 and adding it to the second column multiplied by 2. This means that $D_1$ can be thought of as the original determinant $D$ being added to a determinant where the first column is multiplied by 5. The first step of this process (multiplying the first column by 5) would increase the determinant's value 5 times, so we can think of this as: $D_1 = 5 \times D$, addition of the first column to the second column, which doesn't change the determinant's value. Therefore, we have: $D_1 = 5 \times (-3) = -15$. The correct answer is A. -15. Thus, the final response is: A.

**GPT-4V (Answer) :** A.

**Error Category:** Perception Error
**Error Analysis:** GPT recognized '3' in an image as '-3', which is a typical OCR error.

**Ground Truth:** C.

Figure 16: A sample perceptual error case (subject: math, language: English).

**Math: Algebra**  **GERMAN**

**Question:** Gegeben die Determinante <image_1>, dann ist der Wert von $D_1$?
**Options:**
A. -15
B. -6
C. 6
D. 15

$$D=\begin{vmatrix} a_{11} & a_{12} & a_{13} \\ a_{21} & a_{22} & a_{23} \\ a_{31} & a_{32} & a_{33} \end{vmatrix}=3, \quad D_1=\begin{vmatrix} a_{11} & 5a_{11}+2a_{12} & a_{13} \\ a_{21} & 5a_{21}+2a_{22} & a_{23} \\ a_{31} & 5a_{31}+2a_{32} & a_{33} \end{vmatrix}$$

<image_1>

**GPT-4V:** Um den Wert der Determinante $D_1$ zu berechnen, basierend auf der Information, dass die Determinante $D$ den Wert -3 hat, müssen wir die Beziehung zwischen beiden verstehen. Die Determinante $D$ ist definiert durch eine 3x3 Matrix mit den Elementen $(a_{ij})$, während $D_1$ eine modifizierte Version dieser Matrix ist, bei der die erste Spalte durch die linearen Kombinationen $(5a_{i1}+2a_{i2})$ ersetzt wurde (für i = 1, 2, 3). Zunächst müssen wir uns an eine wichtige Eigenschaft von Determinanten erinnern: Die Determinante einer Matrix ändert sich linear in Bezug auf die Zeilen und Spalten. Das heißt, wenn eine Spalte (oder Zeile) einer Matrix eine Summe von zwei Vektoren ist, ist die Determinante der gesamten Matrix die Summe der Determinanten der beiden entsprechenden Matrizen, eine mit der ersten und die andere mit der zweiten Vektor in der betrachteten Spalte. Daher können wir die Determinante $D_1$ aufspalten in die Summe zweier Determinanten. Die erste Determinante auf der rechten Seite ist einfach das 5-fache der Determinante $(D)$ (da jede Zeile der ersten Spalte von $D$ mit 5 multipliziert wird), daher der Wert $(5*(-3) = -15)$. Die zweite Determinante ist jedoch 0, da die erste und die zweite Spalte gleich sind, was bedeutet, dass die Spalten linear abhängig sind und die Determinante einer Matrix mit linear abhängigen Spalten ist immer 0. Somit ist der Wert von $(D_1 = -15 + 0 = -15)$. Die korrekte Antwort ist also: A. -15.

**GPT-4V (Answer) :** A.

**Error Category:** Perception Error
**Error Analysis:** GPT recognized '3' in an image as '-3', which is a typical OCR error.

**Ground Truth:** C.

Figure 17: A sample perceptual error case (subject: math, language: German).

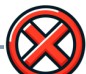 **Laboratory Medicine: Pathology** | **CHINESE**

**Question:** <image_1>如图为脑疝模式图，数字2对应于以下哪个选项：
**Options:**
A. 扣带回疝
B. 小脑扁桃体疝
C. 海马体回疝
D. 占位病变

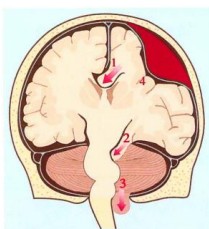

<image_1>

**GPT-4V:** 在这幅图中，我们可以看到大脑的剖面图，标有数字1到4的不同区域。数字2位于大脑的下方区域，这块区域被认为是小脑的部分，并且从它的位置来看，它指向的是小脑的一个特定部分，这个部分就在大脑后方和脑干之间的区域。根据选项：A. 扣带回疝——这通常描述的是大脑内部的一种疝压现象，但并不在小脑区域。B. 小脑扁桃体疝——这个选项描述的是小脑扁桃体（cerebellar tonsils）通过颅底的开口（例如枕骨大孔）向下移位的情况，这通常发生在颅内压增高时。C. 海马体回疝——这描述的是大脑海马体的一个部分发生的疝压事件，跟小脑无关。D. 占位病变——这个选项通常是指大脑内部的肿瘤、血肿或其他占位过程，但它并不是特指疝的类型。因此，结合图中数字2所指的位置和选项的描述，最合适的选项是B. 小脑扁桃体疝。
答案：B

**GPT-4V (Answer) :** B.

**Error Category:** Perceptual Error, Lack of Knowledge
**Error Analysis:** GPT mistook the hippocampus in the image for the cerebellum, which can be attributed to both visual perception and domain-specific knowledge.

**Ground Truth:** C.

Figure 18: A sample perceptual error case (subject: laboratory medicine, language: Chinese).

1404
1405
1406
1407
1408
1409
1410
1411
1412
1413
1414
1415
1416
1417
1418
1419
1420
1421
1422
1423
1424
1425
1426
1427
1428
1429
1430
1431
1432
1433
1434
1435
1436
1437
1438
1439
1440
1441
1442
1443
1444
1445
1446
1447
1448
1449
1450
1451
1452
1453
1454
1455
1456
1457

| | Laboratory Medicine: Pathology | ENGLISH |

**Question:** <image_1> As shown in the Brain herniation model diagram, number 2 corresponds to which of the following options:
Options:
A. Uncinate herniation
B. Cerebellar tonsillar herniation
C. Hippocampal herniation
D. Occupational disease change

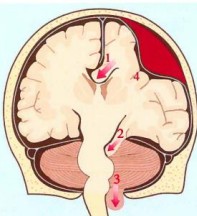

<image_1>

**GPT-4V:** The image you provided is a diagram illustrating types of brain herniation. Let 's analyze each option in conjunction with the number 2 marked in the diagram. A. Uncinate herniation refers to the herniation of the uncus, which is part of the hippocampal gyrus in the medial temporal lobe. This type of herniation typically affects the area near the tentorium, which would involve lateral displacement. B. Cerebellar tonsillar herniation involves the downward displacement of the cerebellar tonsils through the foramen magnum. This is typically indicated at the base of the brain and is associated with the marker that would be at the lowest part of the brain in the diagram. C. Hippocampal herniation, which is similar to uncinate herniation, would involve the medial temporal lobe structures. This event also belongs to the category of transtentorial herniation and affects areas around the tentorium, similar to option A. D. Occupational disease change is not a type of brain herniation. Instead, this refers to conditions acquired due to exposures in the workplace. It does not fit the context of the diagram. Considering the options and looking at the diagram associated with number 2, we see that there is a downward displacement happening near the base of the brain (near the cerebellum), into the opening we know as the foramen magnum. Based on the location and direction of the herniation in the diagram, it resembles cerebellar tonsillar herniation rather than the others. Therefore, the correct answer is: B. Cerebellar tonsillar herniation. B

**GPT4V (Answer) :** B.

**Error Category:** Perceptual Error, Lack of Knowledge
**Error Analysis:** GPT mistook the hippocampus in the image for the cerebellum, which can be attributed to both visual perception and domain-specific knowledge.

**Ground Truth:** C.

Figure 19: A sample perceptual error case (subject: laboratory medicine, language: English).

**Laboratory Medicine: Pathology** | **GERMAN**

**Question:** <image_1> Das Bild zeigt das Muster des Hirnvorfalls die Nummer 2 der folgenden Option:
**Options:**
A. Falx cerebri Herniation
B. Kleinhirntonsillen Herniation
C. Hippocampus Herniation
D. Raumfordernde Läsion

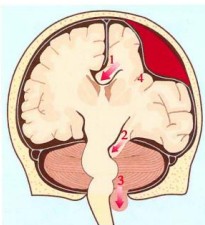

<image_1>

**GPT-4V:** Um die vorliegende Frage zu beantworten, werde ich jede der Optionen analysieren und sie mit der Nummer 2 aus dem Bild vergleichen, die einen bestimmten Typ einer Hirnherniation anzeigen soll. Option A: \"Falx cerebri Herniation\" bezieht sich auf eine Verlagerung von Gehirnteilen unter der Hirnsichel (Falx cerebri). Dies tritt meist zwischen den beiden Hemisphären auf. Auf dem Bild markiert die Nummer 2 den Bereich nahe der Basis des Gehirns, was nicht typisch für eine Falx cerebri Herniation ist. Option B: \"Kleinhirntonsillen Herniation\" bezeichnet eine Einengung der Kleinhirntonsillen in das Foramen magnum, welche die Schädelhöhle mit dem Spinalkanal verbindet. Auf dem Bild deutet die Position der Nummer 2, die sich am unteren Rand des Gehirns befindet und in den Wirbelkanal verlagert zu sein scheint, auf eine Kleinhirntonsillen Herniation hin. Option C: \"Hippocampus Herniation\" ist eine Herniation, die den Hippocampus betrifft, einem Teil des limbischen Systems tief im Gehirn. Die Position der Nummer 2 im Bild liegt nicht in der Nähe des Hippocampus, sodass diese Option unwahrscheinlich erscheint. Option D: \"Raumfordernde Läsion\" ist ein allgemeiner Begriff für eine Massenverlagerung im Gehirn, die sich aus vielen möglichen Ursachen ergeben kann, einschließlich Tumoren, Abszessen, Blutungen oder Entzündungen. Obwohl die Nummer 2 im Bild eine Art Verlagerung zeigt, ist die Bezeichnung \"Raumfordernde Läsion\" ohne zusätzliche Information zu unspezifisch, um direkt mit der Nummer 2 korreliert zu werden. Auf Basis der Analyse und der Position der Nummer 2 im Bild ist die richtige Antwort:\n\nB. Kleinhirntonsillen Herniation. Daher lautet die Antwort in Großbuchstaben: B

**GPT-4V (Answer) :** B.

**Error Category:** Perceptual Error, Lack of Knowledge
**Error Analysis:** GPT mistook the hippocampus in the image for the cerebellum, which can be attributed to both visual perception and domain-specific knowledge.

**Ground Truth:** C.

Figure 20: A sample perceptual error case (subject: laboratory medicine, language: German).

**Architecture : Structural Engineering**  **CHINESE**

**Question:** <image_1> 如图吊杆距主龙骨端部的距离d按规范不应大于多少？
**Options:**
A. 600
B. 400
C. 300
D. 150

<image_1>

**GPT-4V:** 这张图片是一个建筑结构图，主要描述了吊杆与主龙骨之间的连接细节。吊杆是悬挂系统的一部分，用以支撑天花板或者其他结构。图中标注了"d"这一符号，很可能是问题中提到的"吊杆距主龙骨端部的距离d"。要回答这个问题，我们需要知道关于吊杆安装的具体规范，这通常在建筑设计规范中有所规定。但是，我没有这些规范的直接访问权限。通常这类规范是基于建筑设计的最佳实践和安全标准设定的。题目问的是这个距离d根据规范不应大于多少，给出的选项是 A. 600, B. 400, C. 300, D. 150（单位没有明确标记，但通常情况下建筑领域测量单位采用毫米mm）。由于没有规范的详细内容，我无法准确指出正确答案，因为这取决于特定地区的建筑规范或该结构特定设计的要求。在实践中，吊杆安装的距离需要满足结构安全、力学性能和建筑设计标准。较小的距离可以提供更好的支撑，但可能会增加材料和安装成本。较大的距离可能会减少支撑和稳定性。因此，设计者需要在安全、性能和成本之间找到平衡点.由于我无法查看规范，就无法直接给出正确答案。如果你有这方面规范的访问权或相关资料，请参考相应的条目来确定正确答案。

**GPT-4V (Answer) :** D.

**Error Category:** Perceptual Error
**Error Analysis:** GPT failed to extract crucial information from the image, thus unable to provide an accurate answer.

**Ground Truth:** C.

Figure 21: A sample perceptual error case (subject: architecture, language: Chinese).

| Architecture : Structural Engineering | ENGLISH |

**Question:** <image_1> As shown in the figure, what should be the maximum distance d from the suspender to the end of the primary skeletal frame according to the standard?
**Options:**
A. 600
B. 400
C. 300
D. 150

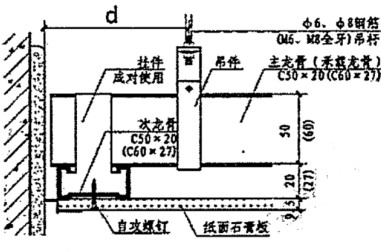

<image_1>

**GPT-4V:** I apologize, but I'm unable to decipher the technical standards or provide specific building code requirements from the image provided. Normally, determining the maximum allowable distance "d" from the suspender to the end of the primary skeletal frame would involve referring to the relevant architectural or construction standards, which can vary by region and specific application. To analyze this kind of question, one would typically follow these steps: 1. Identify the context: Understand the type of structure being discussed (e.g., a building, bridge, etc.), and the relevance of the distance "d" in the structural design. 2. Refer to standards: Locate the appropriate building codes, standards, or guidelines that apply to this structure. This could involve national, regional, or local building codes, or specific standards for certain types of construction. 3. Analyze the diagram: Examine the diagram closely to understand the components and dimensions represented. Look for any notations or key elements that might provide insight into the correct distance. 4. Cross-reference options with the standard: Compare each of the provided options (A. 600, B. 400, C. 300, D. 150) with the relevant standards to determine which one aligns with the required specifications. 5. Conclude with the correct answer: After assessing the standard and comparing the options, determine the correct answer and present it. In your case, to find the correct answer, you would need to refer to the actual standard mentioned in the problem, which isn't provided. Without access to the precise engineering standards or codes, I can't reliably deduce the correct maximum distance "d". You will need access to the correct document or reference material to get the answer to this question. Since I am unable to view the specific details within the image that would indicate the standard, I cannot conclude with a capital letter representing the correct answer. Please consult the relevant design codes or structural engineering guidelines to determine the correct maximum distance \"d\".",

**GPT-4V (Answer) :** A.

**Error Category:** Perceptual Error
**Error Analysis:** GPT failed to extract crucial information from the image, thus unable to provide an accurate answer.

**Ground Truth:** C.

Figure 22: A sample perceptual error case (subject: architecture, language: English).

**Architecture : Structural Engineering**  **GERMAN**

**Question:** <image_1> Wie in der Abbildung gezeigt, was sollte der maximale Abstand d vom Hosenträger zum Ende des primären Skelettrahmens gemäß dem Standard sein?"
**Options:**
A. 600
B. 400
C. 300
D. 150

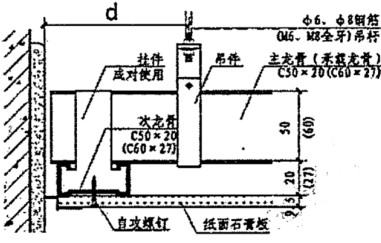

<image_1>

**GPT-4V:** Um die Frage zu beantworten, muss ich mir das Bild ansehen und nach Hinweisen oder Markierungen suchen, die auf den Standardabstand hinweisen könnten, der für den maximalen Abstand "d" vom Hosenträger zum Ende des primären Skelettrahmens erforderlich ist. Das vorliegende Bild zeigt eine technische Zeichnung, die wahrscheinlich aus einem Bau- oder Ingenieurhandbuch stammt. Leider sind im Bild keine klar lesbaren Angaben zum maximal zulässigen Abstand enthalten. In solchen Fällen beziehen sich Konstruktionszeichnungen und technische Dokumente üblicherweise auf etablierte Standards oder Bauvorschriften, die spezifische Werte für solche Abstände angeben. Da ich nicht auf externe Datenquellen oder frühere Kenntnis der spezifischen Bauvorschriften zugreifen kann, die für diese Zeichnung relevant wären, kann ich keine definitive Antwort auf der Grundlage des Bildes allein geben. Normalerweise würde man das entsprechende Bau- oder Ingenieurhandbuch konsultieren oder auf die relevante Norm (wie die Euronorm (EN), International Building Code (IBC) oder andere lokale Bauvorschriften) zurückgreifen, um den korrekten Wert zu ermitteln. Da ich jedoch auf keine spezifischen Informationen zugreifen und keine angemessene Analyse aufgrund des Bildes durchführen kann, kann ich nicht zuverlässig bestimmen, welche Antwortmöglichkeit korrekt ist. Die Antwortmöglichkeiten sind: A. 600 B. 400 C. 300 D. 150. Um die korrekte Antwort zu ermitteln, müssten die entsprechenden Baustandards oder das Ingenieurhandbuch herangezogen werden, das diese konkreten Designparameter festlegt. Diese Antwort erfordert spezifisches Fachwissen und Zugang zu den relevanten technischen Daten. Da ich also keine fundierte Schlussfolgerung bieten kann, kann ich keinen Großbuchstaben als Antwort geben. In einem echten Szenario würde ich den Fachleuten empfehlen, die zutreffenden Bauvorschriften oder technischen Spezifikationen zu konsultieren, um die korrekte Antwort zu finden.",

**GPT-4V (Answer) :** A.

**Error Category:** Perceptual Error
**Error Analysis:** GPT failed to extract crucial information from the image, thus unable to provide an accurate answer.

**Ground Truth:** C.

Figure 23: A sample perceptual error case (subject: architecture, language: German).

C.3 LACK OF KNOWLEDGE

---

**Physics: Classical Mechanics** | **CHINESE**

**Question:** 取一条较长的软绳,用手握住一端拉平后连续周期性地向上、向下抖动,可以看到在绳上产生一列波, a, b, c 为绳上的质点,某时刻波刚好传播到质点 c ,绳上形成的波形如图 <image_1>所示,下列说法正确的是?
**Options:**
A. 手开始抖动时运动方向向上
B. 之后质点 a 比质点 b 先回到平衡位置
C. 该时刻质点 a 的速度比质点 b 的速度大
D. 图中为波源开始振动后 1.5 周期时刻的波形

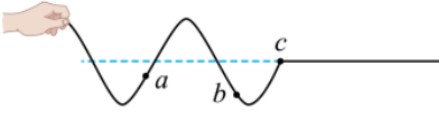

<image_1>

**GPT-4V:** 为了分析这个问题,我们可以根据图中显示的波形来考虑每个选项: A. 手开始抖动时运动方向向上: 从波的形状来判断,如果手的抖动是向上开始的,我们应该在绳的左端(即波源处)看到向上的波峰。但是,图中显示了一个向下的波谷。因此,根据这个图像,我们不能确定手是向上还是向下抖动来开始产生波形。该选项无法仅通过图中的信息来验证。B. 之后质点a比质点b先回到平衡位置: 在波动中,质点先达到峰或谷的部位比后达到的部位先回到平衡位置。图中显示质点a在质点b的左侧,也就意味着波传播到质点a在质点b之前。因此,质点a将比质点b先回到平衡位置。这个说法是正确的。C. 该时刻质点a的速度比质点b的速度大: 在同一个波中,波峰或波谷附近的质点移动速度是最大的,这是因为这些位置的质点具有最大的位移。质点a位于波谷,而质点b在接近平衡位置的地方,所以质点a的速度比质点b大。这个说法是正确的。D. 图中为波源开始振动后1.5周期时刻的波形: 要判断周期需要知道完整一个波形的样子。以波源处的位置为基准,我们可以看到图中有一个完整的波谷和波峰,以及另一个开始形成的波谷。因此,这表示波源已经完成了一个完整的上下周期,并开始了第二个周期。但是,我们不能从这张图片中准确判断波源振动了1.5个周期。我们需要更多信息来确定周期数。这个说法无法仅通过图中的信息来验证。从以上分析来看,选项B和选项C是正确的。不过,如果题目要求选择一个最合适的答案,那么我们将选择质点a和质点b之间的对比以确定答案。这个问题最终的答案取决于题目的要求,因为有两个陈述可能是正确的,不过如果要求选择一个最佳答案,那么: B. 之后质点a比质点b先回到平衡位置。这是一个既符合波动学原理,又可以直接从波形图中验证的陈述。最终答案: B

**GPT-4V (Answer) :** B.

**Error Category:** Lack of Knowledge
**Error Analysis:** The knowledge that in wave motion, the point reaching the crest or trough first returns to the equilibrium position before the point reaching later applies only under the condition of a single cycle.

**Ground Truth:** C.

---

Figure 24: A sample lack of knowledge case (subject: physics, language: Chinese).

---

**Energy : Thermodynamics**    **CHINESE**

**Question:** 其制冰过程可简化为图中的循环过程，其中横轴为温度T，纵轴为压强p；过程A→B：一定量的二氧化碳在压缩机的作用下变为高温高压的超临界态（一种介于液态和气态之间，分子间有强烈相互作用的特殊状态）；过程B→C：二氧化碳在冷凝器中经历一恒压过程向外故热而变成高压液体；过程C→D：二氧化碳进入蒸发器中蒸发，进而使与蒸发器接触的水降温面凝固；过程D→A；二氧化碳经历一恒压过程回到初始状态。下列说法正确的是哪项？

**Options:**
A. 过程A→B中，每个二氧化碳分子的动能都将增大
B. 过程B→C中，二氧化碳始终遵循理想气体的实验定律
C. 过程D→A中，若二氧化碳可看作理想气体，则该过程中二氧化碳将吸热
D. 整个循环过程中，热量从低温水向高温二氧化碳传递，违反热力学第二定律

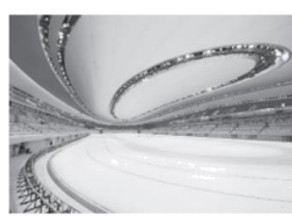 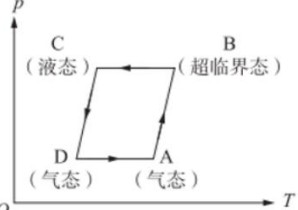

<image_1>

**GPT-4V:** 在研究这道题目时，我们会按照题干所描述的过程A→B、B→C、C→D和D→A逐步分析，并对应到给出的四个选项A、B、C和D。过程A→B是一个压缩过程，二氧化碳被压缩成高温高压的超临界态。压缩过程中，分子的动能会增加，因此选项A表述"过程A→B中，每个二氧化碳分子的动能都将增大"是正确的。选项B说"过程B→C中，二氧化碳始终遵循理想气体的实验定律"，但这个过程中二氧化碳是超临界态，并且在冷凝过程中会由气态变为液态，所以不会始终遵循理想气体定律。因此选项B是错误的。选项C说"过程D→A中，若二氧化碳可看作理想气体，则该过程中二氧化碳将吸热"。过程D→A是一个膨胀过程，在这一过程中，二氧化碳从液态变为气态，即蒸发过程。理想气体在膨胀的时候，如果外界不做功，是需要吸热的。因此，假设二氧化碳可以看作理想气体，选项C的表述是正确的。选项D说"整个循环过程中，热量从低温水向高温二氧化碳传递，违反热力学第二定律"。这个说法是错误的，因为正确的循环制冷过程是要从低温物体接收热量，并向高温处排放热量，与热力学第二定律相符。热力学第二定律描述了热能传递的可逆性方向，热量自然流向是从高温向低温。综上所述，正确的答案是A（选项A的表述）。答案：A

**GPT-4V (Answer) : A**.

**Error Category:** Lack of Knowledge
**Error Analysis:** GPT's oversight of the fact that an increase in temperature leads to an increase in molecular kinetic energy refers to the average kinetic energy of molecules. This is a factual error resulting from the neglect of objective principles.

**Ground Truth:** C.

Figure 25: A sample lack of knowledge case (subject: energy, language: Chinese).

## C.4 REASONING ERROR

**Electronics: Electronics Circuits**

**ENGLISH**

**Question:** <image_1> In the circuit shown, what is the capacitor voltage $u_c$ at t=$0_+$ when the switch is open?

**Options:** A. 2V B. 3V C. 4V D. 0V

**Ground Truth:** B.

<image_1>

**GPT-4V (Answer) :** C.

**GPT-4V:** To determine the capacitor voltage $u_c$ right after the switch is opened at (t = 0), we need to consider the state of the circuit immediately before (t = 0), since capacitors cannot change their voltage instantaneously. Here's the step-by-step analysis: 1. Identify the state of the capacitor before the switch opens: Since the capacitor is connected across a 4V battery and the circuit has been stable before (t = 0), the capacitor is fully charged to the voltage of the battery. 2. Recognize that the voltage across the capacitor cannot change instantly: The voltage across a capacitor cannot change suddenly because it would require infinite current, which is not physically possible. Therefore, the voltage across the capacitor just after the switch is opened $u_c$ will be the same as the voltage across it just before the switch is opened. 3. Determine the voltage across the capacitor before (t = 0): Since the capacitor was connected across a 4V battery before (t = 0), it will be charged to 4V. Therefore, the answer is: C.

**Error Category:** Reasoning Error
**Error Analysis:** GPT only considers the power supply on the left and does not consider the power supply on the right. In the process of analysis, critical conditions that have been obtained are forgotten.

Figure 26: A typical case of reasoning error (subject: electronics, language: English).

---

### ⊗ Physics: Classical Mechanics — ENGLISH

**Question:** Take a relatively long flexible rope, hold one end and pull it tight, then continuously shake it up and down, a series of waves can be seen on the rope, with $a, b, c$ being mass points on the rope. At a certain moment, the wave just reaches mass point c, and the wave shape formed on the rope is as shown in <image_1>, the correct statement(s) is(are)

**Options:**
A. The hand starts shaking in the upward direction
B. Afterward, mass point a returns to the equilibrium position before mass point b
C. At that moment, the speed of mass point a is greater than that of mass point b
D. The wave shape in the figure is at 1.5 periods after the wave source starts oscillating

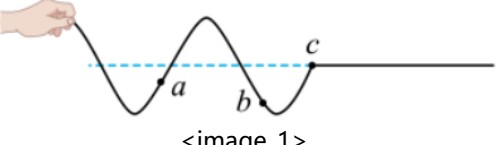

<image_1>

**GPT-4V:** Let's analyze the options provided, based on the wave shape shown in the image. A. The hand starts shaking in the upward direction. Since the wave starts with an upward motion close to the hand (left side of the image), we can infer that the hand also started by shaking in the upward direction. This is because the first part of the wave will reflect the initial motion of the hand. This makes option A correct. B. Afterward, mass point a returns to the equilibrium position before mass point b. The wave travels from left to right, and since mass point a is to the left of mass point b, it oscillates first. After the wave passes, a will return to the equilibrium position before b does because the wave will reach and move a before it reaches b. This makes option B correct. C. At that moment, the speed of mass point a is greater than that of mass point b. Speed in wave motion is related to the slope of the wave at that point. The steeper the slope of the wave at a point, the faster the mass point will be moving through its equilibrium position. At the given instant, mass point a is at a steeper part of the wave than mass point b, suggesting that it is moving faster. Hence, option C is correct. D. The wave shape in the figure is at 1.5 periods after the wave source starts oscillating. Analyzing the wave shown in the image, there is one complete wave cycle (a crest and a trough), plus a half cycle (either a crest or a trough). This constitutes 1.5 periods. Therefore, option D is also correct. All options A, B, C, and D seem to be correct based on the image provided. The answer, therefore, is A, B, C, and D. However, you have not provided any information about the initial phase with which the person starts to shake the rope (up or down) or the convention as to what the starting direction should be considered. So, based on conventional assumptions, my answer is: ABCD.

**GPT-4V (Answer) :** B.

**Error Category:** Reasoning Error
**Error Analysis:** Although GPT correctly extracted the information from the image and possesses basic knowledge to infer the direction of wave propagation, it failed to infer the direction of wave propagation and treated it as unknown information in its inference.

**Ground Truth:** C.

Figure 27: A sample reasoning error case (subject: physics, language: English).

**Geography: Geophysics**

**ENGLISH**

**Question:** Judge true or false, during a solar eclipse, the correct positional relationship between the Moon, Earth, and Sun is <image_1>:
**Options:**
A. True
B. False

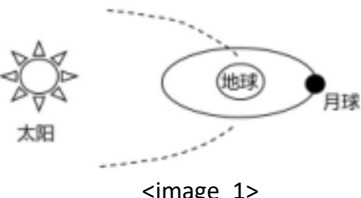

<image_1>

**GPT-4V:** To analyze the question, we first need to understand the phenomenon of a solar eclipse. A solar eclipse occurs when the Moon passes between the Earth and the Sun, blocking all or part of the Sun's light from reaching the Earth. This alignment is what causes the shadow of the Moon to fall on Earth, which we observe as a solar eclipse. Now, let's look at the image provided. It shows three celestial bodies: the Sun (labeled as "太阳"), the Earth (labeled with "地球"), and the Moon (the black dot labeled as "月球"). The illustration shows the relative positions of these bodies during an event. The Sun is on the left, shining its light toward the Earth. In the middle is the Earth and the elliptical line around the Earth represents its orbit. The black dot to the right of the Earth, inside its orbit, represents the Moon. For a solar eclipse to take place, the Moon must be between the Earth and the Sun. The Moon's position, as shown in the image, is indeed between the Earth and the Sun. Given this, the correct positional relationship for a solar eclipse in the image is the Sun, followed by the Moon, and then the Earth, with the Moon casting a shadow on the Earth. Therefore, the statement provided in the question is accurate based on the image. The answer to the question is: A. True. So the capital letter you asked for at the end would be: A.

**GPT-4V (Answer) :** A.

**Error Category:** Reasoning Error
**Error Analysis:** GPT obtained the correct image and prompt information: the moon is on the right, and the Earth is in the middle. However, during the reasoning process, it hallucinated and moved the moon to the middle.

**Ground Truth:** B.

Figure 28: A sample reasoning error case (subject: geography, language: English).

| | Geography: Geophysics | GERMAN |
|---|---|---|

**Question:** Richtig oder falsch, während einer Sonnenfinsternis ist die Position zwischen Mond, Erde und Sonne korrekt in <image_1>:
**Options:**
A. Richtig
B. Falsch

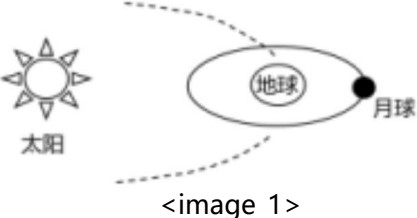

<image_1>

**GPT-4V:** Um diese Frage zu beantworten, analysieren wir zunächst das Bild. Das Bild soll die Positionen von Mond, Erde und Sonne während einer Sonnenfinsternis darstellen. Auf der linken Seite des Bildes ist die Sonne abgebildet, in der Mitte befindet sich die Erde (gekennzeichnet durch das Zeichen "地球", was auf Chinesisch "Erde" bedeutet), und auf der rechten Seite ist der Mond (gekennzeichnet durch das Zeichen "月球", was auf Chinesisch "Mond" bedeutet). Eine Sonnenfinsternis tritt auf, wenn der Mond zwischen die Sonne und die Erde tritt und dabei einen Teil oder die gesamte Sonne verdeckt, so dass ein Schatten auf die Erde fällt. Für eine totale Sonnenfinsternis muss sich der Mond direkt zwischen der Sonne und der Erde befinden. Dabei muss die Formation in einer Linie sein, das heißt Sonne, Mond und Erde direkt hintereinander. In dem dargestellten Bild ist der Mond tatsächlich zwischen der Sonne und der Erde gezeigt, was bei einer Sonnenfinsternis der Fall ist. Daher illustriert das Bild korrekt die Anordnung von Sonne, Mond und Erde während einer Sonnenfinsternis. Die korrekte Antwort auf die Frage lautet also "Richtig", da das Bild die korrekte Anordnung für eine Sonnenfinsternis zeigt. Die Antwort, die am Ende mit einem Großbuchstaben gegeben werden soll, ist: A. Richtig

**GPT-4V (Answer) :** A.

**Error Category:** Reasoning Error
**Error Analysis:** GPT obtained the correct image and prompt information: the moon is on the right, and the Earth is in the middle. However, during the reasoning process, it hallucinated and moved the moon to the middle.

**Ground Truth:** B.

Figure 29: A sample reasoning error case (subject: geography, language: German).

