# OpenReview forum: "M4U: Evaluating Multilingual Understanding and Reasoning for Large Multimodal Models"
_ICLR.cc/2025/Conference — ICLR 2025 Conference Withdrawn Submission_

### Official Review · Reviewer_unB7 · 2024-11-02

**Soundness:** 2
**Presentation:** 1
**Contribution:** 2
**Rating:** 3
**Confidence:** 4

**Summary:**

This paper introduces M4U, a novel and challenging benchmark for evaluating multilingual multimodal understanding and reasoning capabilities across 16 subfields in Science, Engineering, and Healthcare in three languages. The experiments show that even the state-of-the-art model, GPT-4, still has significant room for improvement, especially in expert-level multilingual multimodal reasoning.

**Strengths:**

1. This paper introduces a carefully annotated multimodal, multilingual reasoning dataset, consisting of 8,931 multiple-choice questions covering 64 disciplines across 16 subfields in Science, Engineering, and Healthcare.
2. The paper tests various MLMMs on this dataset to evaluate their performance on subject-specific multimodal reasoning and analyzes the causes of their limitations.

**Weaknesses:**

1. Necessity of the benchmark. It is suggested to provide specific examples of real-world applications or use cases where this benchmark would be valuable for evaluating MLLMs. Emphasizing its importance in areas like educational robots or medical scenarios might help clarify its necessity.
2. General applicability of the benchmark. This benchmark requires extensive subject-specific knowledge, making reasoning skills secondary. The emphasis on subject knowledge could limit its general applicability. It is suggested to reorganize the title and the Introduction section to highlight specific scenarios and necessary requirements for the benchmark. For example, in MMMU(MMMU: A massive multi-discipline multimodal understanding and reasoning benchmark for expert AGI), the title emphasizes the scenario is "expert AGI".
3. Limited Contribution. The main focus of the paper is on introducing the benchmark and evaluating various MLLMs on it. However, there is no novel approach in terms of the design, construction, or evaluation methods of the benchmark, nor is there an attempt to improve upon methods that performed poorly. This makes the contribution seem limited.

**Questions:**

1. The benchmark content is quite challenging. Given the difficulty level, it’s hard to envision which specific models or fields would absolutely need this benchmark.

2. While M4U offers a more refined, discipline-specific benchmark, it may not serve as a true innovation in assessing multimodal understanding and reasoning, as it primarily tests subject-specific knowledge. The need for such specialized knowledge might be limited outside of subject-focused educational contexts. Narrowing the scope to specific application areas—such as healthcare, education, or scientific research—would help make the benchmark’s significance and applicability more compelling.

Therefore, I suggest you provide evidence or examples demonstrating how M4U tests multimodal understanding and reasoning capabilities beyond just subject knowledge. These examples would help to clarify the benchmark's value and innovation.

3. As shown in the analysis in Figure 4, the top performance barriers are “Perceptual Error”, “Lack of Knowledge” and “Reasoning Error” with the first two accounting for over 69%. This suggests that the main performance limitations stem from the MLLM’s image encoder and LLM lacking subject-relevant knowledge, rather than reasoning capability. Therefore, describing this dataset as a measure of reasoning and understanding might be somewhat misleading.

It is suggested to provide additional analyses or examples that demonstrate how the dataset tests reasoning capabilities despite the prevalence of perceptual and knowledge-based errors. You can show examples of those failures due to “Reasoning Error”. These examples are the characteristics of your benchmark.

4. In Line 112-113, "Unlike MMMU and CMMMU, our dataset focuses on the evaluation of multilingual multimodal reasoning". As far as I know, MMMU and CMMMU are also multi-science and multi-modal benchmarks. The difference is that your benchmark can evaluate problems in multiple languages, and the problems are harder.

I wonder if the multilingual aspect truly necessary? While it is essential in certain contexts, this point isn’t clearly presented. Additionally, Chinese and German are not particularly low-resource languages. It’s unclear why multilingualism is a primary focus in this work.

5. It is suggested to detail the process of the data collection and annotation processes. A high-quality dataset requires not only accurate labels but also well-defined annotation standards and quality control procedures. This information is essential for assessing the dataset’s reliability and usefulness, as well as aiding other researchers in replicating and leveraging the data. The paper should include details about annotator selection, annotation processes, quality control measures, and how label consistency and accuracy were ensured. For example, the following could be expanded: “To minimize the risk of data contamination, samples are collected from college exams and quizzes from online video lectures. Additionally, a significant portion (35%) of the questions in M4U are written by our team based on textbooks” (line 82-84). For the first part, what type of crowdsourcing platform was used? For the latter, how did you extract questions from textbooks, formulate questions, and choose answer options? Highlighting the technical aspects of crowdsourcing and data engineering might make this work more engaging.

---

### Official Review · Reviewer_uoLL · 2024-11-03

**Soundness:** 2
**Presentation:** 3
**Contribution:** 3
**Rating:** 5
**Confidence:** 4

**Summary:**

This paper proposes a new challenging benchmark for assessing the capabilities of LMMs in multi-discipline, multilingual, and multimodal understanding and reasoning. The benchmark covers various disciplines and languages, and the authors have meticulously collected and constructed this benchmark. The authors conducted extensive evaluations of leading LMMs and LLMs, demonstrating that there is still significant room for improvement in existing models' capabilities in multilingual multimodal reasoning.

**Strengths:**

-  The motivation behind this benchmark is well-founded, the benchmark provides support for subsequent evaluations of the capabilities of LMMs/LLMs in multidisciplinary, multilingual, and multimodal understanding and reasoning.
- The construction of the benchmark is thorough, the paper offers a complete assessment and analysis of LMMs and LLMs.
- The paper is well-written and easy to understand.

**Weaknesses:**

- The data sources for the benchmark are mainly focused on science exam questions, and there is still room for improvement in terms of comprehensiveness.
- Compared to other benchmarks, there does not seem to be a significant difference.
- The evaluation method is relatively simple and may not fully reflect the model's multilingual understanding and reasoning abilities.

**Questions:**

- Were more evaluation methods attempted, such as translating questions into English or incorporating OCR results of images into the questions, to compare the models' understanding and reasoning abilities?

---

### Official Review · Reviewer_jdRQ · 2024-11-04

**Soundness:** 2
**Presentation:** 2
**Contribution:** 2
**Rating:** 3
**Confidence:** 4

**Summary:**

The paper begins by collecting examination questions with relatively low leakage risk from Chinese universities, resulting in a Chinese multimodal assessment dataset comprising 8,931 items. The authors then constructed English and German translation assessment datasets by translating the text only. They claim that this dataset is a particularly challenging one for multimodal large models and explore the cross-linguistic performance of these models.

**Strengths:**

Through extensive labor in collection and processing, the authors have created a relatively large multimodal reasoning assessment dataset that reflects university-level knowledge. This effort addresses the issues of high data leakage risk in existing datasets and the inability to differentiate performance among various models.

**Weaknesses:**

> 1. Exaggeration of Contributions:

The article's scientific contributions focus on "challenging visual reasoning tasks" and "multilinguality." However, it lacks sufficient experimentation and effort to substantiate and showcase these contributions.

- Regarding the contribution of "challenging visual reasoning tasks," the authors attempt to demonstrate this by collecting university-level questions with low leakage risk and arguing that these tasks are "difficult" from the perspectives of "answering based solely on text" and "low model accuracy." However, they do not provide data on human expert performance for these questions, which undermines their definition of "difficulty" due to the lack of reference to human accuracy rates.

- Concerning the "multilingual" contribution, merely translating text into two other languages does not convincingly support the claim of "multilinguality," especially in light of the absence of experimental data to ensure the accuracy of the translation, casting doubt on the quality of the multilingual aspect of the dataset.

- Most importantly, as described in the paper, if the images contain Chinese text, retaining the Chinese in the assessments of other languages is clearly unreasonable. Furthermore, considering that most multimodal models' visual encoders are not trained on Chinese text-image pairs, their ability to encode Chinese text is limited, significantly constraining the models' reasoning capabilities and potentially leading to inaccurate assessments of the dataset.

> 2. Lack of Rigor and Missing Information in Data Processing:

For the critical data quality verification aspect of the dataset, the authors use vague language. For instance, they repeatedly refer to "over 10 college and graduate students" when discussing the number of annotators, which I believe is an inadequate expression for a scientific paper. Additionally, there are other missing data statistics, as detailed in the Questions section below.

**Questions:**

Q1: How is the accuracy of the multilingual data translation ensured? Were native speakers involved in verifying the accuracy?

Q2: Why was the translation of text in the images not considered?

Q3: The paper mentions that the dataset questions come from online videos, PDFs, and textbooks. What are the respective proportions of these sources?

Q4: How is the accuracy of the questions generated ensured? Was there a cross-checking process, and what is the consistency rate of the statistics?

---

### Official Review · Reviewer_GZnn · 2024-11-05

**Soundness:** 3
**Presentation:** 3
**Contribution:** 3
**Rating:** 6
**Confidence:** 4

**Summary:**

This paper curates an evaluation benchmark (called M4U) for multilingual multimodal models. It consists of 8,931 samples covering 64 disciplines across 16 subfields in Science, Engineering, and Healthcare in Chinese, English, and German. The evaluation shows that it's a challenging benchmark for most LMMs, where GPT-4o achieves only 47.6% average accuracy on M4U.

**Strengths:**

* **Writing**: The paper is well-written and easy to follow.
* **Well-designed Curation Process**: A large portion of the dataset is curated to avoid data contamination.
* **Contribution**: It would be a useful resource for the community to assess the multilingual performance of LMMs.

**Weaknesses:**

* **Language Coverage**: Only three languages (Chinese, English, and German) are covered in this benchmark.

* **Language-specific characteristics**: The multilingual samples are directly translated from one language therefore the language-specific characteristics are not considered in the curation process. Especially for STEM, it's very much evaluating the translation ability or the multilingual ability of the model. It would be interesting to cover samples where the interesting characteristics (e.g., culture) behind multilingualism really matter. For example, for healthcare, the guideline in different countries may differ so having those knowledge is important and will make the benchmark more interesting.

**Questions:**

* Is there any language-specific samples covered in the current dataset?

* Is it possible to evaluate only the OCR ability of different models to probe if these models fail because of the multilingual OCR ability?

---

### Note · Authors · 2024-11-24

I have read and agree with the venue's withdrawal policy on behalf of myself and my co-authors.